# Resolving Domain Shift For Representations Of Speech In Non-invasive Brain Recordings

## Abstract

Machine learning techniques have enabled researchers to leverage neuroimaging data to decode speech from brain activity, with some amazing recent successes achieved by applications built using invasive devices. However, research requiring surgical implants has a number of practical limitations. Non-invasive neuroimaging techniques provide an alternative but come with their own set of challenges, the limited scale of individual studies being among them. Without the ability to pool the recordings from different non-invasive studies, data on the order of magnitude needed to leverage deep learning techniques to their full potential remains out of reach. In this work, we focus on non-invasive data collected using magnetoencephalography (MEG). We leverage two different, leading speech decoding models to investigate how an adversarial domain adaptation framework augments their ability to generalize across datasets. We successfully improve the performance of both models when training across multiple datasets. To the best of our knowledge, this study is the first ever application of feature-level, deep learning based harmonization for MEG neuroimaging data. Our analysis additionally offers further evidence of the impact of demographic features on neuroimaging data, demonstrating that participant age strongly affects how machine learning models solve speech decoding tasks using MEG data. Lastly, in the course of this study we produce a new open-source implementation of one of these models to the benefit of the broader scientific community.

## 1 Introduction

Applications leveraging recent advancements to decode representations of speech in the brain stand to positively impact the lives of individuals across the world who suffer from impaired verbal communication. While surgically invasive modalities provide the most direct access to the brain, they are practically and ethically prohibitive to conduct at scale. Thus, researchers have increasingly turned towards non-invasive approaches instead. However, non-invasive modalities also come with a unique set of challenges, including a difficult signal-to-noise ratio.

We choose to focus on magnetoencephalography (MEG) as our neuroimaging modality of interest and speech decoding as the principle class of the objectives for the models we train. Specifically, we look at heard speech (listening to someone else speak) decoding as this field is still in its infancy and it is easier to decode than imagined speech (thinking intently of what one is saying without vocalizing it) (Martin et al., 2014). This choice is supported by evidence, albeit contested (Vicente & Langland-Hassan, 2018), of functional overlap between the neural representations of heard and imagined speech (Wandelt et al., 2024). We select MEG because it sits at the intersection between many of the advantages of other non-invasive techniques. While functional magnetic resonance imaging (fMRI) has a stronger spatial resolution than MEG, it is limited in its temporal resolution. Both MEG and electroencephalography (EEG), on the other hand, can record activity on the milisecond timescale at which the brain operates (Hall et al., 2014). Yet in comparison to EEG, MEG has superior spatial resolution, a higher signal-to-noise ratio and a far greater number of scalp-based sensors on average (Hall et al., 2014). In addition, there is evidence to suggest that the use of MEG over EEG is directly correlated with increased performance for speech decoding (Défossez et al., 2023).

Overall, the decoding performance of applications using non-invasive modalities continues to lag behind invasive ones - with one reason being the limited scale of the data in most non-invasive studies. Despite efforts to acquire increasingly large datasets and curate open neural data repositories, the field has not been able to recreate the the successes that deep learning and "big data" have seen elsewhere. This is due, in large part, to the fact that non-invasive neuroimaging data is inherently difficult to generalize from. For one, different studies employ a myriad of scanners and task designs (Jayalath et al., 2024). Pooling data across scanners and sites then leads to an increase in non-biological variance caused by the differences in the devices and acquisition, including scanner manufacturer (Han et al., 2006)(Takao et al., 2014), upgrade (Han et al., 2006), drift (Takao et al., 2011), strength (Han et al., 2006), and gradient nonlinearities (Jovicich et al., 2006). Additionally, within any given study, participants exhibit anatomical and demographic differences that affect the signals recorded from their brains (Jayalath et al., 2024). Providing a reliable means of overcoming this hurdle is an active area of research for both the neuroscience and computer science communities. While data harmonization is generally the preferred term among neuroimaging researchers, among computer scientists this problem is most commonly characterized as dataset bias or domain shift (Gretton et al., 2008). In the literature, these two terms are used interchangeably to refer to the same phenomenon. In this study, we present the first application of feature-level harmonization to address domain shift for MEG neuroimaging data. We demonstrate the relevance of this framework for speech decoding by improving the ability of two different networks to generalize across datasets during training to increase performance.

## 2 RELATED WORK

Domain adaptation (DA) is one approach for solving the domain shift problem which comes from the family of Transfer Learning methods. The bulk of the literature focuses on unsupervised domain adaptation (UDA) as it is a more challenging task that can be trivially adapted for the supervised case. In general, techniques for solving UDA can be categorized as either statistic moment matching (e.g. Long et al. (2018)), domain style transfer (e.g. Sankaranarayanan et al. (2018)), self-training (e.g. Zou et al. (2020); Liu et al. (2021)), or feature-level adversarial learning (e.g. Ganin et al. (2016); He et al. (2020a;b); Liu et al. (2018)) (Liu et al., 2022). Domain shift is often measured by the dissimilarity of the distributions of each domain. A number of metrics have been proposed to this end (Ben-David et al., 2006; 2010; Mansour et al., 2009; 2012; Germain et al., 2013), but the notion most relevant to present study is that of $\mathcal{H}$-divergence. Based on the work of Kifer et al. (2004), it was later used in the formalization of Ben-David et al. (2006; 2010)'s theory on domain adaptation. This same theoretical framework led to the Domain-Adversarial Neural Networks (DANN) architecture, one of the first successful deep approaches for DA (Ganin et al., 2016). Inspired by generative adversarial networks (GANs), Tzeng et al. (2017) extended the idea behind DANNs with their Adversarial Discriminative Domain Adaptation (ADDA) architecture. An extension of this line of work to $N$ source dimensions was subsequently demonstrated by (Zhao et al., 2019).

Given the importance of removing dataset bias (particularly scanner-induced variance) for neuroimaging studies, it is no surprise that a large number of studies have tasked themselves with resolving domain shift in this area. Much of the existing work is based on an empirical Bayes method called ComBat (Johnson et al., 2006). However, ComBat is primarily applied to image-derived values and associations (which MEG is not). The literature in this area has thus focused largely on structural, functional, and diffusion MRI. In fact, DA has been explored more for diffusion MRI than any other modality - the drawback being many of the methods produced rely on spherical harmonics, limiting the ability to apply them to other neuroimaging techniques (Dinsdale et al., 2021). A few deep approaches have been tried, such as leveraging variational autoencoders (VAEs) (Moyer et al., 2020) and generative models based on the U-Net (Ronneberger et al., 2015) or cycleGAN (e.g. Dewey et al. (2019); Zhao et al. (2019)) architectures. However, these methods are sometimes limited by inherent difficulties validating the harmonized outputs that are generated (Dinsdale et al., 2021). Very few studies to date have leveraged Ben-David et al.'s theoretical UDA framework in the context of neuroimaging data and, to our knowledge, none with MEG data.

An exception to this is the work by Dinsdale et al. (2021). Building on the body of work around $\mathcal{H}$-divergence, they show that an ADDA-style framework (Tzeng et al., 2017) can be successfully adapted to improve cross-dataset generalization for MRI data. We will refer to this ADDA-style approach as *adversarial harmonization* throughout the remainder of this work. Specific to MEG

data, Jayalath et al. (2024) introduce an alternative approach that also manages to find success leveraging data from multiple studies. They propose a pre-training scheme that demonstrates cross-task and cross-dataset generalization wherein the combinations of data used across pre-training and fine-tuning encompass different datasets, each employing distinct scanner types and task designs (Jayalath et al., 2024). However, this cross-dataset generalizability remains limited in its efficiency for leveraging aggregated MEG data at the scale required for deep learning. We thus select the architecture proposed by Jayalath et al. (2024) as one of the two base models we investigate for improving MEG cross-dataset generalization. The second architecture we examine was proposed by Défossez et al. (2023) and reports strong results training over individuals pooled from a single study. Despite showing their model's performance scales with the number of individuals used during training, they do not report a further attempt to train their architecture over multiple datasets. For the sake of convenience, we often refer to these two models by the name of their original code repositories: Brainmagick, for Défossez et al. (2023), and MEGalodon, for Jayalath et al. (2024).

## 3 METHODS

### 3.1 DATASETS AND PREPROCESSING

This work focuses on four MEG datasets across the two architectures it extends. The Cambridge Centre for Ageing and Neuroscience (Cam-CAN) data repository (Shafto et al., 2014; Taylor et al., 2017) contains 641 subjects covering 160 hours of MEG recordings in total. Armeni et al. (2022) contains 30 total hours of recordings (three subjects each listening to 10 hours of speech) and (Gwilliams et al., 2022)'s Manually Annotated Sub-Corpus (MEG-MASC) dataset contains 54 hours (27 subjects each recorded for 2 hours). Lastly, Schoffelen et al. (2019)'s Mother Of Unification Studies (MOUS) consists of 204 subjects recorded for a calculated total of 160 hours. Differences in the devices used during acquisition can carry such a strong signal in the final data that the terms *dataset bias* and *scanner bias* are sometimes used interchangeably. However, because the MEG scanner types used are not mutually exclusive among the studies we examine, we make a particular choice to focus on the term dataset bias in a way that is inclusive of, but broader than, acquisition device and configuration. We refer to these datasets by the names of their primary authors (i.e. Gwilliams) or their monickers (i.e. MEG-MASC) throughout the rest of this work. Table 1 summarizing the above information is included for the reader's convenience.

| Primary Author | Monicker | Scanner Brand | MEG Hours |
|---|---|---|---|
| Armeni et al. (2022) | - | CTF | 30 |
| Gwilliams et al. (2022) | MEG-MASC | KIT | 54 |
| Shafto et al. (2014), Taylor et al. (2017) | Cam-CAN | Elekta Neuromag | 160 |
| Schoffelen et al. (2019) | MOUS | CTF | 160 |

Table 1: A reference table of the relevant dataset information. Total data volume of each dataset is reported in hours.

Noting the impact of different demographic distributions between neuroimaging datasets on harmonization found by Dinsdale et al. (2021), we examine the normalized distributions of both participant age (see Figure 1) and participant sex (see Figure 5 in the Appendix) for each pair of datasets used during training. The Brainmagick experiments leverage the Gwilliams and MOUS datasets, while the MEGalodon experiments use the MOUS and Cam-CAN datasets. We construct a set of subsets both to ease constraints on computational resources as well as examine demographic effects. The Gwilliams and MOUS datasets have relatively equivalent age and sex distributions and therefore no specific measures need to be taken to normalize these features when constructing subsets. While there is also no significant disparity related to the ratios of participant sex between the MOUS and Cam-CAN datasets, we do find a large difference when examining the distributions of participant age. To this end, we construct two different pairs of subsets for these datasets. The first selects individuals at random only from the area of overlap between the two age distributions. We refer to these as *balanced subsets*. The second set of subsets, which we call *random subsets*, selects individuals randomly from each dataset such that they approximate the distribution of participant age from the original studies. An even split of males and females overall is controlled for in both cases. In all

cases, approximately 15 percent of subjects from each dataset are used in total to ensure the ratio of total subjects and MEG recording hours between any two datasets is maintained. The demographic visualizations relating to each class of subset can be found in Section A.3 of the Appendix.

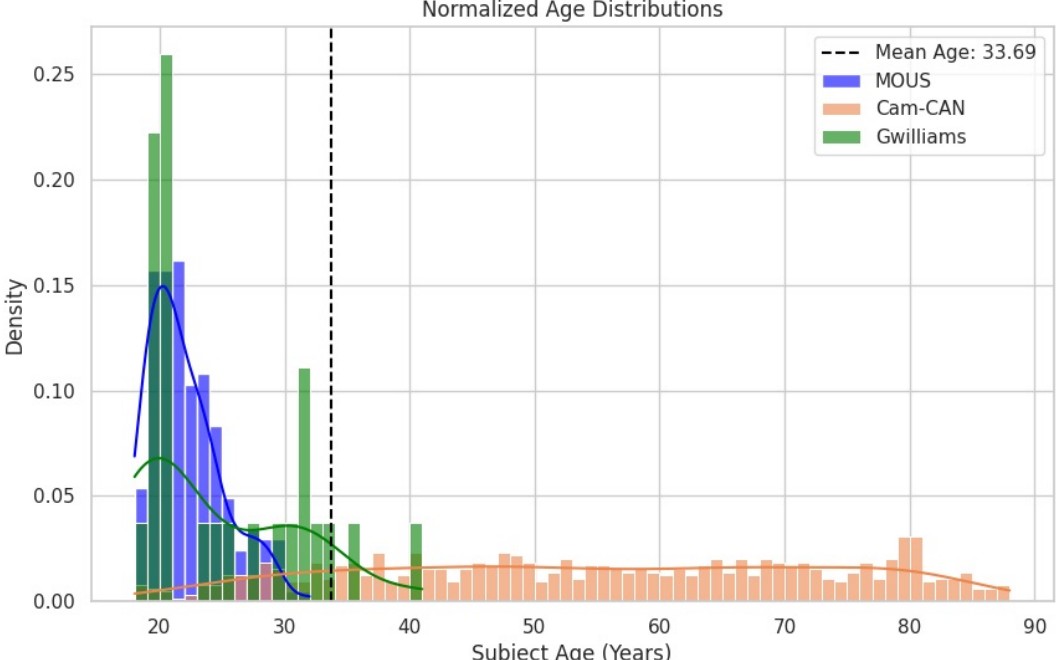

Figure 1: The normalized distributions of the ages of the subjects from the MOUS Schoffelen et al. (2019), Cam-CAN Shafto et al. (2014); Taylor et al. (2017), and Gwilliams et al. Gwilliams et al. (2022) datasets. The density plotted along the y-axis represents the proportion (i.e. relative frequency) of each category within its respective dataset. The mean age calculated over the three datasets is displayed by the dotted line.

We focus on harmonizing the deep representations of speech in the brain at the feature level, and as a result apply only whatever minimal preprocessing of MEG data the original papers when implementing our chosen architectures. The full scope of the preprocessing steps carried out are detailed in Section A.2 of the Appendix.

## 3.2 DECODING TASKS

For the experiments building off of Défossez et al. (2023)'s Brainmagick model, the decoding task is to predict directly the most probable segment of speech stimulus from the corresponding period of MEG data. For the experiments using Jayalath et al. (2024)'s MEGalodon model, the pre-text objectives are a set of domain-specific classification tasks proposed in the original paper: band prediction, phase shift prediction, and amplitude scale prediction. The speech decoding tasks selected for the fine-tuning phase are speech detection and voicing classification. The goal for speech detection is to determine whether speech has occurred in the section of continuous auditory stimulus given the corresponding segment of MEG data. This should not be confused with the more trivial task of detecting the onset of speech from rest. In voicing classification, phonemes must be classified as voiced or voiceless from the aligned MEG data.

## 3.3 ADVERSARIAL HARMONIZATION

An architecture augmented with adversarial harmonization is generally composed of a feature extractor (which we will refer to as the encoder block), a label predictor (which we will refer to as the task head), and a domain classifier. All parts of the network are first trained together to convergence in a 'warm up' phase to ensure both that the encoder block is able to produce salient features and

that the domain classifier is able to distinguish between them accurately. In the harmonization or 'unlearning' phase, the network is then trained via an iterative training procedure composed of three steps: (1) optimizing the encoder and task head for the primary task, (2) optimizing the domain classifier to identify the remaining dataset bias, and (3) optimizing the encoder to remove the dataset bias by confusing the domain classifier (Dinsdale et al., 2021). Each of these steps optimizes a unique loss function and as (2) and (3) are adversarial in nature, they cannot be updated concurrently. Thus, the result is three iterations for each training batch.

Each of the passes during the adversarial phase of the harmonization framework requires its own optimizer, including the initial warm-up phase, for a total of four optimizers used. We keep the original choice of AdamW (Loshchilov & Hutter, 2019) for control epochs (with all parameters updated by a single optimizer) and at first followed the lead of Dinsdale et al. (2021) in using Adam (Kingma & Ba, 2017) for the remaining optimizers. However, hyperparameter testing revealed that the use of a stochastic gradient descent (SGD) optimizer for the adversarial domain classifier led to smoother training during harmonization for both the domain classifier and encoder. The idea to examine different optimizers was informed by the work of Rangwani et al. (2022) who formalize the idea of smoother convergence through the use of SGD when training the adversarial head. This work further confirms their theory over a new domain. A comparison of the performance during training is shown in Section A.8 of the Appendix. For the work involving the MEGalodon framework we retain the learning rate of 0.000066 from the original study for the warm-up phase and follow Dinsdale et al. (2021) in using a learning rate of 0.00001 during harmonization. Alternative choices for the second learning rate are explored, but no real effect is found. Similarly, for the work related to the Brainmagick base model we retain the original paper's learning rate of 0.0003 for the warm-up phase and before reducing it to 0.00001 for all optimizers during the adversarial stage.

## 3.4 BRAINMAGICK EXPERIMENTS

For the experiments using Défossez et al. (2023)'s architecture as a base, we begin by training the proposed CLIP model first using the original code released alongside the paper and then again with our own implementation of the base model. Leveraging the existing implementation proved to be an unanticipated challenge, in part due to the use of the Python libraries Flashy and Dora developed internally for Facebook Research and relied upon extensively in their code for the Brainmagick paper. These tools have limited available documentation and are not widely used by other research groups outside of Facebook Research. One major outcome of the present study is therefore simply the creation of an open-source implementation of the Brainmagick architecture relying on the standard Pytorch[1] and Lightning[2] Python libraries. We then look to continue the work of the original authors and investigate whether their architecture benefits from dataset pooling with minimal alterations. We explore a version of "naive" pooling via a pre-training scheme where we train on each dataset, leveraging the saved weights from a previous training run on the other. For example, we train the model on the MOUS dataset except loading the best (as determined by validation loss) saved weights of the baseline run on the Gwilliams dataset. Next, we train the model from random initialization with the same splits but pooling all of the data. Finally, we look to explore whether augmenting the network with adversarial harmonization offers a boost to performance. In all cases we use a 0.7 : 0.1 : 0.2 train/val/test split.

The Brainmagick model is implemented faithfully to the original design, though with steps taken to streamline and simplify the repository as a whole - including a critical bug fix related to sensor labeling. The spatial attention layer is used to transform data from both the Gwilliams and MOUS datasets to a uniform number of output channels (270) allowing for the datasets to be processed together. This layer was reported by the authors as being originally designed to support a cross-dataset model, as working with multiple studies requires the ability to generalize over different numbers and locations of sensors. As in Défossez et al. (2023), we use a value of 0.2 for the dropout component. The remainder of the *brain model* remains true to the original design. In the case of adversarial harmonization, we treat the brain model as the *encoder block*. The model used alongside the CLIP loss forms the *task head*, with the CLIP loss remaining as the task loss for harmonization. Lastly, we add our *domain classifier* at the same level as the task head. Following the iterative training regime established by Tzeng et al. (2015), we use cross entropy loss for the

---

[1] https://pytorch.org/
[2] https://lightning.ai/docs/pytorch/stable/

domain classifier and employ the confusion loss first proposed by Tzeng et al. (2015) and leveraged by Dinsdale et al. (2021) in optimizing the encoder block to erase the target bias from our deep feature representations. We offer further details and formal definitions of spatial attention, the CLIP loss, and the confusion loss in Section A.4 of the Appendix. The augmented architecture can be found there in Figure 9 as well. All related code is available in this repository[3].

## 3.5 MEGalodon Experiments

The backbone of Jayalath et al. (2024)'s architecture is a dataset-conditional layer (which projects all MEG recordings into a shared dimensional space) and cortex encoder (which extracts deep representations of brain activity). Additionally, we choose to follow the original authors in applying the optional subject embeddings to the final output of the backbone. When pre-training, these features have a projection applied to them before being used to solve a series of "pre-text" tasks. If fine-tuning, the output of the encoder is used directly for the speech decoding tasks. In applying adversarial harmonization, we treat all layers up to and including the optional subject conditioning as the encoder block. Similarly, the areas responsible for the pre-text and fine-tuning objectives are grouped as the task head. At the same level as the task head, we then add a domain classifier that forms the adversarial component of our harmonization scheme. We also keep the task loss from the original MEGalodon framework for both phases of harmonization (warm-up and adversarial). As before, we use cross entropy loss and the confusion loss for each backward pass involving the domain classifier. Further details on the original MEGalodon architecture and an illustration of the augmented architecture are given in Section A.5 and Figure 10 of the appendix, respectively.

The MEGalodon pre-text tasks, by design, apply some transformation to the input and then pass this transformed input through the backbone in order to create a set of transformed features to be used for the actual prediction task. However, because the effect is running the entire encoder and creating a unique set of features one time for each task, we pass each additional feature vector through the domain classifier as well. The losses from each of these are aggregated by summation before performing the backwards pass. As we use three pre-text tasks, every training step handles four total feature vectors. We set the value of $\alpha$, the scaler variable applied to the cross-entropy loss of the domain classifier, to 0.25 to account for this.

In the case where participant age is targeted instead of dataset bias, we again follow the harmonization roadmap laid out by Dinsdale et al. (2021). In order to adapt the continuous feature of age to a categorical one such that it can be predicted by the adversarial classifier, we create 72 single-year bins spanning from the youngest age across all the datasets (18) to the oldest (89). However, it is also important to capture the fact that for a true age of 25, a prediction of 24 is more accurate than a prediction of 63. To account for this, both the true age labels and the predicted ages (produced by applying a softmax activation to the output of the classifier and then taking the argmax) are converted to softmax labels normally distributed as a $\mathcal{N}(\mu, \sigma^2)$ where $\mu$ is equal to the discrete age value and $\sigma$ is set to 10. Lastly, the cross-entropy loss is swapped out in favor of the Kullback-Leibler (KL) divergence, where we treat the distance of the softmax distribution of the predicted ages from the softmax distribution of the true ages as the loss value. All code related to the augmented MEGalodon architecture is available in this repository[4].

## 4 Results

### 4.1 Brainmagick Results

The re-implementation of the Brainmagick architecture proposed by Défossez et al. (2023) using more widely documented libraries is a success. Our version performs comparatively to the results reported in the paper, albeit with a slight reduction in performance which we attribute to our choice to train all the runs related to our build on a single GPU. As the original authors note in their repository[5], the number of GPUs used during training can have a large impact when using contrastive losses and for this reason we use the single GPU results of the control runs of our build as our primary

---

[3]https://anonymous.4open.science/r/BMBU-9C3E/README.md
[4]https://anonymous.4open.science/r/megalodon-harmonizer-22A3/README.md
[5]https://github.com/facebookresearch/brainmagick

| Full-Run Results | | Top-10 Accuracy | |
| --- | --- | --- | --- |
| Method | Training Data | Gwilliams | MOUS |
| Control (Official repo) | Gwilliams | 70.7%, 70.7%* | - |
| Control (Official repo) | MOUS | - | 68.5%, 67.5%* |
| Control (Our implementation) | Gwilliams | 69.8% | - |
| Control (Our implementation) | MOUS | - | 68.1% |
| Pre-trained on MOUS | Gwilliams | 68.8% | - |
| Pre-trained on Gwilliams | MOUS | - | 67.1% |
| Control | Gwilliams + MOUS | $68.8\% \pm 0.5$ | $66.8\% \pm 0.4$ |
| Harmonized | Gwilliams + MOUS | $\textbf{71.0}\% \pm 0.2$ | $\textbf{68.6}\% \pm 0.2$ |

Table 2: Following the convention of Défossez et al. (2023), we report Top-10 segment-level accuracy with confidence intervals calculated over 3 seeds. Results as reported in the original study are denoted by a single asterisk (*). The best performance recorded over each validation dataset is marked in bold.

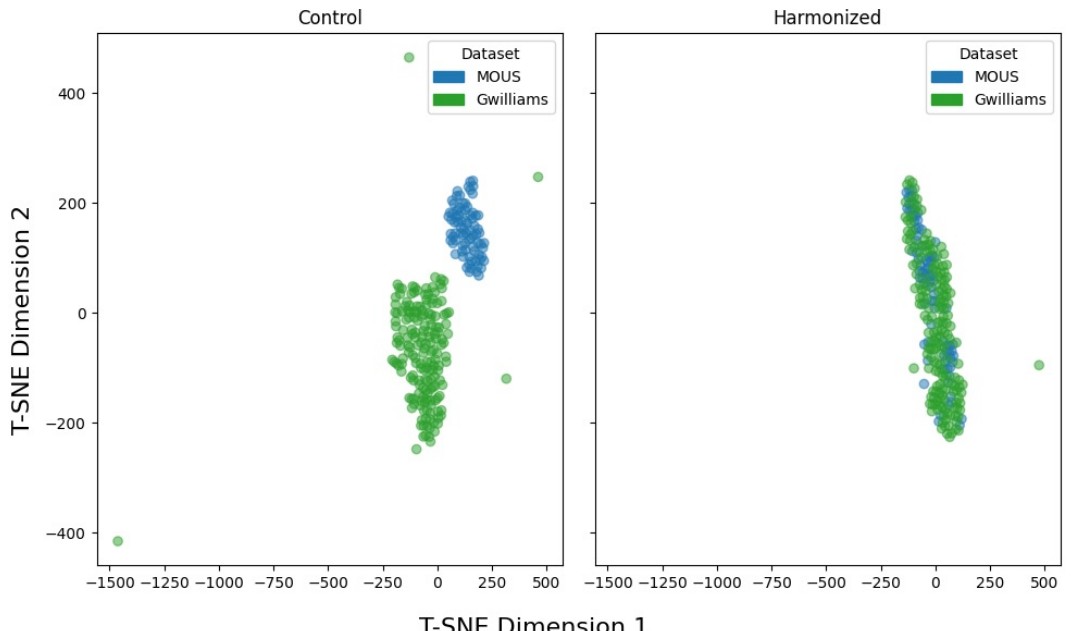

Figure 2: t-SNE plot (van der Maaten & Hinton, 2008) of the activations of the final layer of the encoder block using the Brainmagick (Défossez et al., 2023) base architecture while training over subsets. The left plot shows the control and the right after harmonization. The control data is approximately linearly separable, while the harmonized data is closely mixed.

baseline comparison. We do not find that the base Brainmagick architecture is effective in cross-dataset generalization, as performance decreases when using both MOUS and Gwilliams during training. This was true for both the attempt at naively combining the datasets via a pre-training approach as well as training on the datasets pooled together directly. However, we find that using adversarial harmonization yields a 2.2% increase in performance evaluating over the Gwilliams test split and 1.8% increase in performance for the MOUS test split when pooling datasets for training. We conduct a one-sided independent samples $t$-test using the results collected across three seeds and find that our augmentations are statistically significant ($p < 0.05$) for both the Gwilliams test split ($p = 0.012$) and MOUS test split ($p = 0.011$). In fact, adversarial harmonization allows the architecture to successfully combine the datasets during training to improve top-10 accuracy even over the results reported in the original paper. All results from training with the full datasets are shown in Table 2. These findings are additionally supplemented by analogous results collected over subsets of the data (see Table 4 in the Appendix). Together, these results demonstrate a scaling effect as overall data volume is increased.

| Armeni Fine-Tuning | | Balanced Accuracy | |
|---|---|---|---|
| Method | Pre-training Data | Speech Detection | Voicing |
| Control | Balanced (M+CC) | 57.29% | 52.60% |
| Control | Random (M+CC) | 56.53% | 52.38% |
| Warm-up Only | Balanced (M+CC) | **57.76**% | 52.35% |
| Short Warm-up (dataset) | Balanced (M+CC) | 56.33% | 51.99% |
| Harmonized (dataset) | Balanced (M+CC) | 55.04% | 52.44% |
| Harmonized (dataset) | Random (M+CC) | 56.25% | 52.42% |
| Harmonized (age) | Random (M+CC) | 50.68% | 50.82% |
| Harmonized (both) | Random (M+CC) | 56.15% | **52.65%** |

Table 3: We report the balanced accuracy results for the speech detection and voicing classification tasks, fine-tuning and testing on the Armeni dataset. Balanced refers to subsets with no strong bias related to the age distribution of the participants with respect to either dataset, while for Random this is not controlled for. The confound being harmonized is denoted in parentheses in the method column. All runs reported here are trained for 200 epochs. Short Warm-up denotes pre-training the encoder for 100 epochs without the domain classifier, before training with the classifier for an additional 10 epochs and beginning harmonization at epoch 110. Warm-up only indicates training in the warm-up phase for the entire 200 epochs.

We further support our claim of cross-dataset generalization by observing that upon beginning the harmonization phase, the dataset classifier is reduced from an average 99.9% accuracy to an average 79.7% and 67.9% accuracy in the full and subset cases, respectively. Additionally, we extract the features produced by the final layer of the encoder block and visualize the change in the separability of the activations through a t-SNE plot (van der Maaten & Hinton, 2008) shown in Figure 2.

### 4.2 MEGalodon Results

The results support our hypothesis that the skewed demographic features of the MOUS and Cam-CAN datasets are a possible cause of the difficulty Jayalath et al. (2024)'s model has attempting to scale the number of datasets used during pre-training. We show that the performance of the original model is improved for both decoding tasks when training with the age-balanced subsets as opposed to the random subsets (see Table 3). We conduct experiments just on the age-balanced subsets to determine the degree to which dataset of origin exists as a confound independent of age distribution. Even in this case, a randomly initialized dataset classifier achieves 99.9% accuracy after only a single epoch of training. The features produced by the pre-training scheme therefore have dataset-identifiable aspects beyond those related to participant age. Augmenting the model with adversarial harmonization, we successfully manage to lower the dataset classification accuracy to 51% on average (a reduction of 48.9%). Using the random subsets and targeting age bias, we see in Figure 4 that the softmax probability distribution of the classifier for participant age is driven closer to universal chance after training with adversarial harmonization. Fine-tuning results for dataset harmonization, age harmonization, and jointly harmonizing for both age *and* dataset bias are also shown in Table 3.

We find that pre-text task validation loss is a direct proxy for speech decoding performance in the case of speech detection, but they become uncoupled for voicing classification. Harmonization has a negative effect on speech detection performance in all cases, yet jointly harmonizing for both age and dataset bias delivers better voicing classification performance than the control on both the random *and* age-balanced subsets - despite having a much worse final pre-text task validation loss at the end of training. t-SNE (van der Maaten & Hinton, 2008) plots comparing the final encoder block activations from the end of the warm-up phase to the end of harmonization are shown in Figure 3.

## 5 Discussion

The increase in fine-tuning performance for the MEGalodon architecture when using the age-balanced subsets compared to the random subsets (Table 3) demonstrates that the demographic features of a subject strongly affect the characteristics of the data collected using MEG devices.

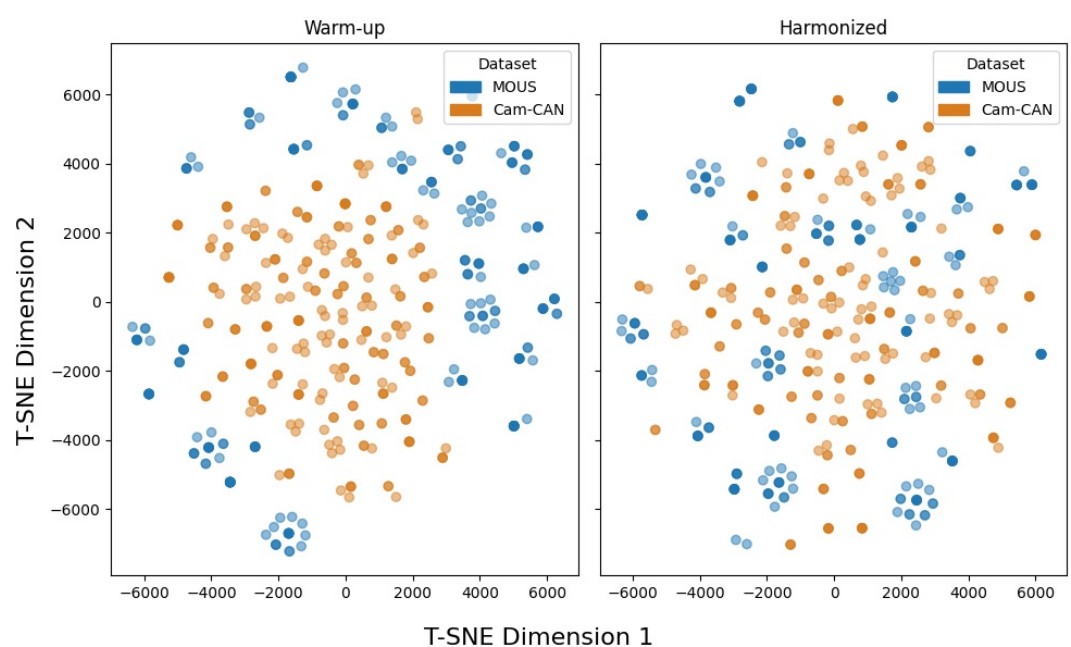

Figure 3: t-SNE plots van der Maaten & Hinton (2008) of the activations of the final layer of the encoder block of the base MEGalodon Jayalath et al. (2024) architecture when harmonizing jointly for dataset and age bias. The plot on the left shows the results at the end of the warm-up phase, while the right shows the results after harmonization. Training was completed on the random subsets.

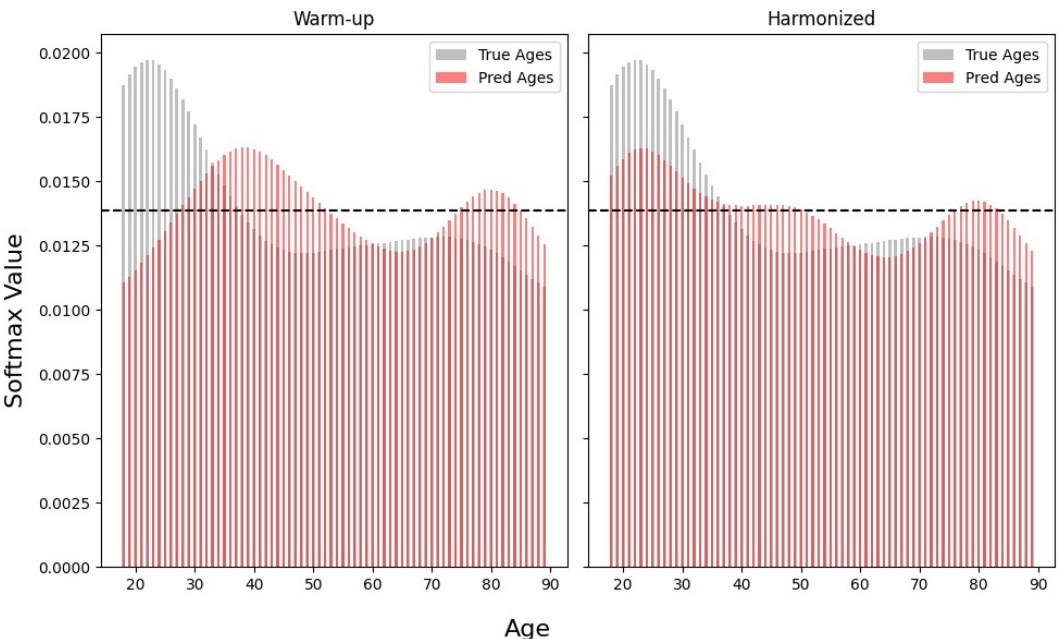

Figure 4: Softmax values of the true and predicted ages averaged over a single batch and converted into Gaussian distributions. The dashed line represents the value of setting all softmax values equal. The plot on the left shows the output of the model at the end of the warm-up phase, while the one on the right shows the output after an additional 100 epochs of adversarial harmonization. After harmonization, the distribution is flattened towards an equal distribution across all ages.

However, the particular direction of this effect is likely due to the fact that the datasets used for evaluation (whether that be Armeni or Gwilliams) both have distributions of participant age much more similar to that of the MOUS dataset. While the Cam-CAN dataset includes individuals ranging from 18 years old to 89 years old, the other three datasets don't record any participants older than 41, with most younger than 30. Were the fine-tuning datasets to have more inclusive age-distributions, as is the case with Cam-CAN, the results of the control comparison might be flipped. This indicates a broader need within the neuroimaging community to increase efforts to recruit older participants when conducting these studies. Still, the results found here indicate that harmonization methods could play a critical role in reducing age-related effects in both present and future studies.

Additionally, we find adversarial harmonization can be extremely unstable, with task loss diverging sharply when the harmonization phase begins. This effect can be mitigated for dataset bias through hyperparameter selection. As the models are still able to reduce the task loss after beginning harmonization, the dataset-identifying features present in the encoder output must not be necessary to perform speech decoding. We were not able to complete equivalent hyperparameter testing for the experiments targeting age as a confound. While the harmonized Brainmagick model was able to improve validation loss over the control, the MEGalodon model could not within our training horizon. We believe one of the differentiating factors between these two architectures is the speed at which they reach convergence. This is examined in further detail, including related runtime and batch size experiments, in Section A.9 and A.8 of the Appendix.

Lastly, as we note above, augmenting the MEGalodon architecture with adversarial harmonization has a negative effect for speech detection and a positive one for voicing classification. This is likely due to the difference in the fine-tuning protocol established by Jayalath et al. (2024) between the tasks rather than a significant qualitative difference. 'Shallow' fine-tuning the network for speech detection only updates the task head, but both the encoder and task head are updated when 'deep' fine-tuning for voicing classification. The discrepancy in performance between the decoding tasks could indicate that harmonization drives the features into a more universal brain representation but at the initial cost of task-specific (in this case *task* referring broadly to all speech perception) performance. This is resolved in the deep fine-tuning case as the encoder is given the chance to re-focus on the downstream task, but remains salient for shallow fine-tuning as the encoder is kept frozen.

## 6 IMPACT AND FUTURE WORK

To the best of our knowledge, this study is the first ever application of feature-level, deep learning based harmonization for MEG neuroimaging data. We demonstrate some of the unique challenges of harmonizing MEG data when compared to other modalities and demonstrate that age-related features strongly affect how machine learning models solve speech decoding tasks from MEG data. Using the Brainmagick and MEGalodon architectures as a base, we achieve success augmenting the ability of two different, leading speech decoding models to generalize between datasets. We produced these results even without extensive hyperparameter testing, meaning there were likely performance gains still left on the table.

A continuation of this work involving an exhaustive hyperparameter search for both models and unrestricted training time for the augmented MEGalodon model would be well-warranted, as it could not be completed within the scope of the present study. Future work should also explore how the effects reported here hold when pooling training data from upwards of three datasets. We acknowledge the limitations of this study in the Appendix. However, the scope of the work produced remains significant and we believe this study to be of value to the field. As a whole, the results reported here are evidence for the potential of adversarial harmonization to aid in solving the scaling problem for deep learning applications when it comes to MEG data.

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

# A APPENDIX

## A.1 COMPUTE

As all experiments were carried out on a shared resource used by multiple research groups, a single GPU was used in order to reduce wait times. Additionally, limitations related to specific configurations during the period of this project's completion meant full-run experiments on the MEGalodon architecture were not feasible to carry out. As it stands, even relying on subsets approximately 15% the size of the full datasets for the vast majority of our experiments, we estimate to have used over 1,600 hours of GPU compute in the completion of this study.

## A.2 ADDITIONAL PREPROCESSING DETAILS

All work using the MEGalodon framework as a base applied standard procedures using the native functionality of the MEG dataloaders from the PNPL python library[6]. A low-pass filter was applied at 125 Hz and high-pass filter at 0.5 Hz in order to remove artifacts from muscle movements and slow-drift, respectively. Additionally, a notch filter is applied at multiples of 50 Hz to account for possible line noise from the electric grid where the original recordings were taken. The signal is also downsampled to 250 Hz, taking care to avoid aliasing at frequencies up to the threshold set by the low-pass filter. Bad sensor channels are then detected with a variance threshold and replaced by interpolation from the nearest sensors (Jayalath et al., 2024). The Brainmagick framework has unique requirements in the way data is sliced, batched, and tracked and therefore is not able to rely

---

[6]https://github.com/neural-processing-lab/pnpl

on the PNPL library. Specifically, corresponding MEG and speech stimulus data were segmented into 3 second windows and stimulus segments were tracked to enforce no overlap between train, validation, and test splits. In addition to applying a scaler (implemented via the scikit-learn python library (Pedregosa et al., 2012)), the average over the first 5 seconds was taken and subtracted from each channel as a baseline correction for signal artifacts. Finally, the data was normalized and values greater than 20 standard deviations were clamped to minimize outliers (Défossez et al., 2023).

## A.3   DEMOGRAPHIC VISUALIZATIONS

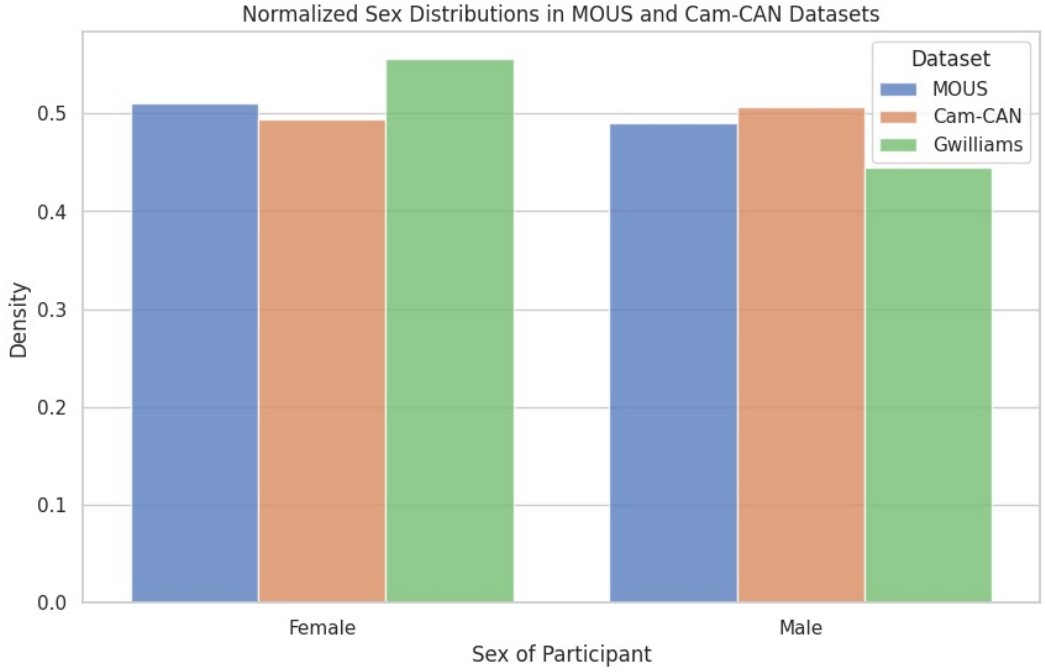

Figure 5: The normalized distributions of the sex of the subjects from the MOUS (Schoffelen et al., 2019), Cam-CAN (Shafto et al., 2014; Taylor et al., 2017), and Gwilliams et al. (2022) datasets. The density plotted along the y-axis represents the proportion (i.e. relative frequency) of each category within its respective dataset.

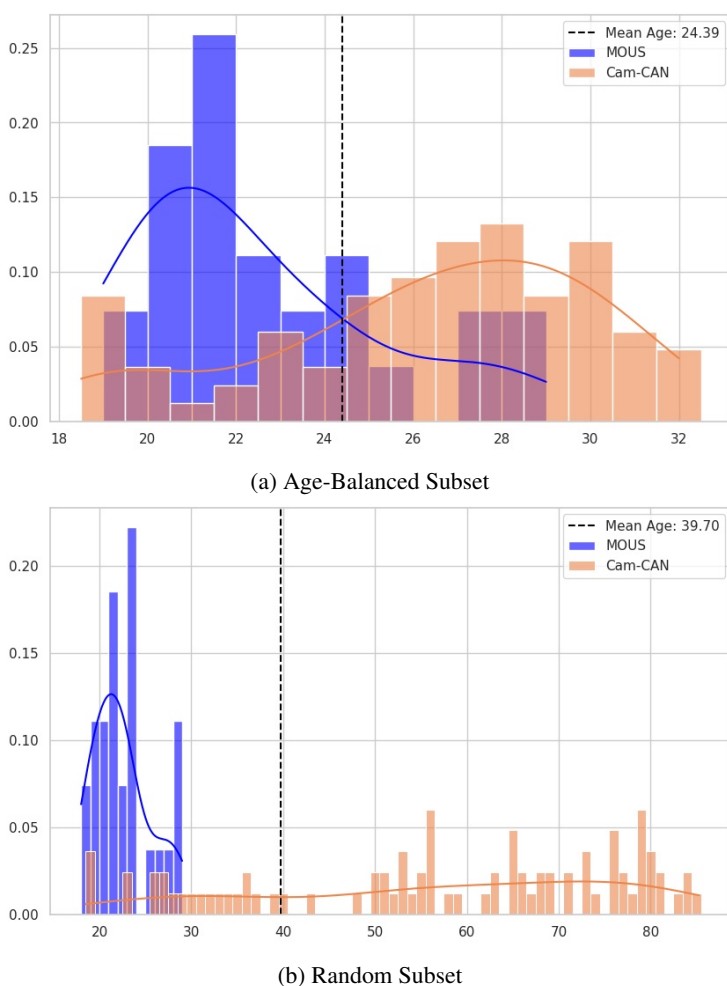

(a) Age-Balanced Subset

(b) Random Subset

Figure 6: The normalized distributions of participant ages from subsets taken over the MOUS (Schoffelen et al., 2019) and Cam-CAN (Shafto et al., 2014; Taylor et al., 2017) datasets for experiments done with the MEGalodon (Jayalath et al., 2024) base architecture. The density plotted along the y-axis represents the proportion (i.e. relative frequency) of each category within its respective dataset. The mean age calculated over the two datasets is displayed by the dotted line. The age-balanced subsets were created by randomly selecting subjects from the overlap of the two whole dataset age distributions. The random subsets include subjects taken at random from the entire distributions. In both cases the count of Male and Female participants is balanced with a tolerance of 1 subject in either direction.

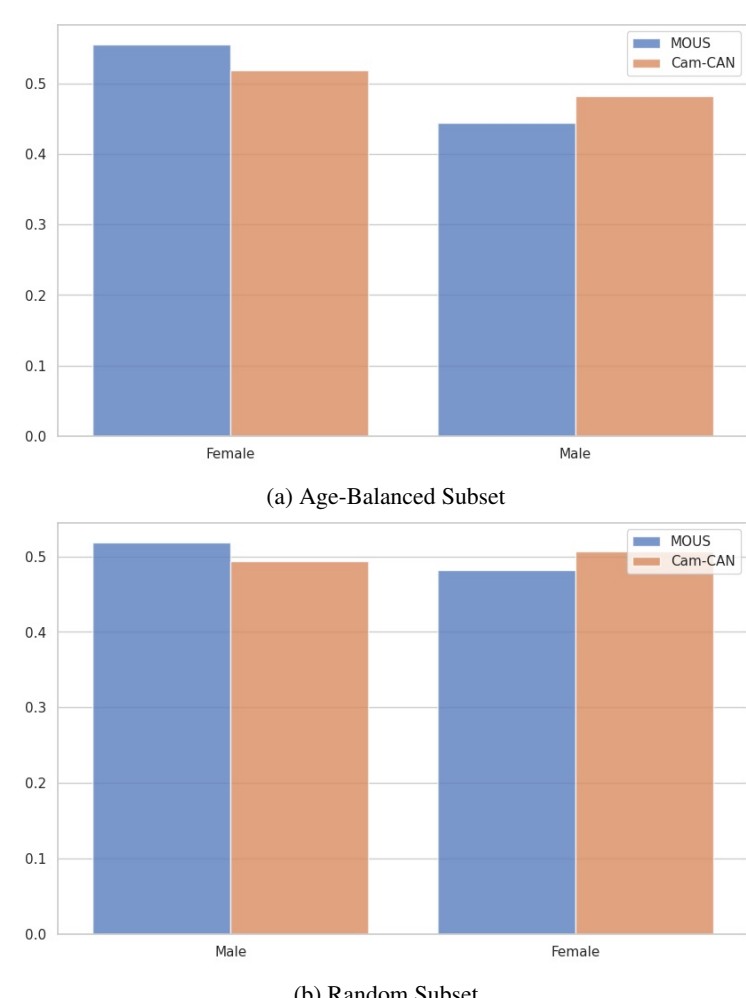

(a) Age-Balanced Subset

(b) Random Subset

Figure 7: The normalized distributions of participant sex from subsets taken over the MOUS (Schof-felen et al., 2019) and Cam-CAN (Shafto et al., 2014; Taylor et al., 2017) datasets for experiments done with the MEGalodon (Jayalath et al., 2024) base architecture. The density plotted along the y-axis represents the proportion (i.e. relative frequency) of each category within its respective dataset. The age-balanced subsets were created by randomly selecting subjects from the overlap of the two whole dataset age distributions. The random subsets include subjects taken at random from the entire distributions. In both cases the count of Male and Female participants is balanced with a tolerance of 1 subject in either direction.

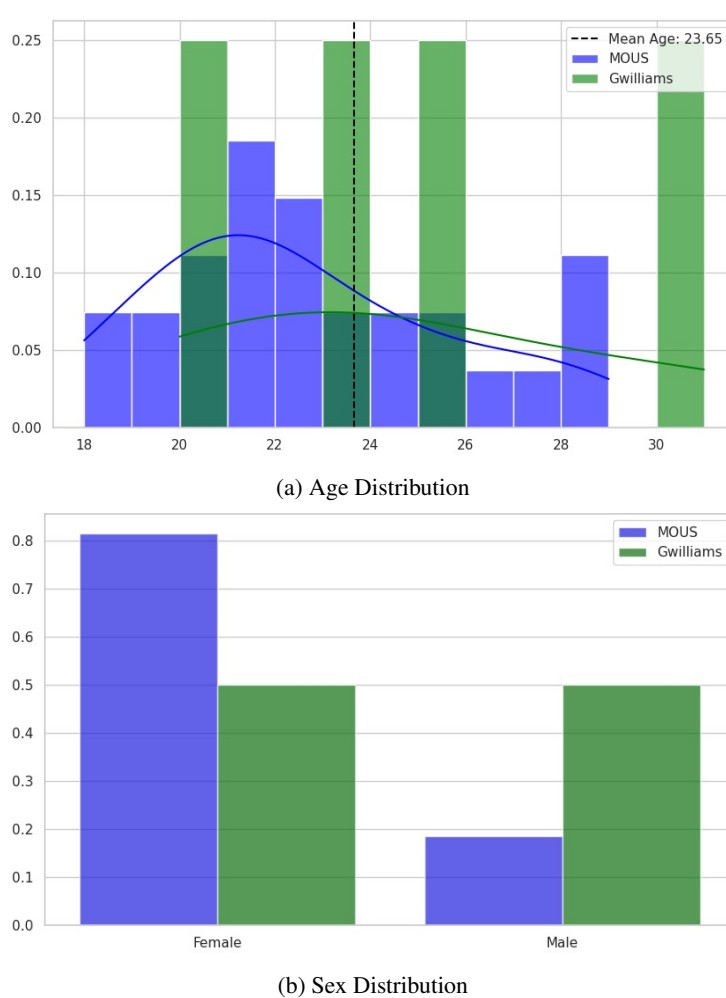

(a) Age Distribution

(b) Sex Distribution

Figure 8: The normalized distributions of participant ages and participant sexes from subsets taken over the MOUS (Schoffelen et al., 2019) and Gwilliams et al. (2022) datasets for experiments done with the Brainmagick (Défossez et al., 2023) base architecture. The density plotted along the y-axis represents the proportion (i.e. relative frequency) of each category within its respective dataset. The mean age calculated over the two datasets is displayed by the dotted line. Subsets were constructed with the intention of mimicking the characteristics of the full distributions from which they were sampled.

## A.4 ADDITIONAL BRAINMAGICK DETAILS

Défossez et al. (2023)'s architecture is composed generally of a speech module, a brain module, and a contrastive loss. Wav2Vec 2.0 (Baevski et al., 2020), a self-supervised model trained on audio alone, is used for the speech module as they find it best represents the latent representations of speech sounds (Défossez et al., 2023). The brain module is constructed sequentially from a spatial attention layer over the MEG (or EEG) sensors, a participant ($1 \times 1$ convolution) layer, and a set of convolutional blocks (Défossez et al., 2023).

### A.4.1 SPATIAL ATTENTION

The spatial attention layer helps select the most salient sensors from the layouts used by different studies during collection when remapping the MEG data into a shared channel dimension. The design works by first projecting the three-dimensional sensor locations (i.e. input channels), $i$, to a two-dimensional plane. This is done using a function from the MNE (Gramfort et al., 2013) Python library that leverages a device-dependent surface meant to preserve channel distances. These two-

dimensional positions $(x_i, y_i)$ are then normalized to $[0, 1]$ and for each output channel, $j$, a function $a_j$ over $[0, 1]^2$ is learnt. This function is parameterized in the Fourier space as $z_j \in \mathbb{C}^{K \times K}$ with $K$ = 32 harmonics along each axis, giving the full function definition as

$$a_j(x, y) = \sum_{k=1}^{K} \sum_{\ell=1}^{K} \text{Re}(z_j^{(k,\ell)}) \cos(2\pi(kx + \ell y)) + \text{Im}(z_j^{(k,\ell)}) \sin(2\pi(kx + \ell y)). \tag{1}$$

The final weights over the input sensors are found by taking the softmax of the function $a_j$ evaluated at the sensor locations $(x_i, y_i)$:

$$\forall j \in \{1, \ldots, D_1\}, \text{SA}(X)^{(j)} = \frac{1}{\sum_{i=1}^{C} e^{a_j(x_i, y_i)}} \left( \sum_{i=1}^{C} e^{a_j(x_i, y_i)} X^{(i)} \right) \tag{2}$$

where SA is the spatial attention (Défossez et al., 2023). Because $a_j$ is periodic in practice, $(x, y)$ are scaled down and a spatial dropout is applied by sampling a location and removing each sensor within a specified distance from the softmax.

### A.4.2 CLIP Loss

Défossez et al. (2023)'s Brainmagick architecture uses a multi-modal CLIP (originally Contrastive Language-Image Pre-Training) loss (Radford et al., 2021). Commonly, a regression loss is used in the supervised training of decoders to predict latent representations of speech known to be relevant to the brain - in many cases the Mel spectrogram due to its similarity to how sound is represented in the cochlea (Mermelstein, 1976). The problem with regression objectives in this context, however, is that they rely on the assumption that the dimensions of the Mel spectrogram are all scaled correctly and equally important. In reality, some (e.g. very low) frequencies are irrelevant to speech and can be differentiated by irregular orders of magnitude. The CLIP loss importantly does not aim to maximally distinguish speech segments from one another but acts to relax the constraints of a regression loss which may be tied too heavily to the above assumptions of relevancy, accuracy, and scaling with respect to the representations from the speech module (Défossez et al., 2023). Given a brain recording segment $X$ and the representation of the corresponding speech sound $Y \in \mathbb{R}^{F \times T}$, $N - 1$ negative samples $\tilde{Y}_j \in \{1, \ldots, N - 1\}$ are taken over the dataset and a positive sample is added as $\tilde{Y}_N = Y$. The training objective therefore becomes predicting the probability $\forall j \in \{1, \ldots, N\}, p_j = \mathbb{P}\left[\tilde{Y}_j = Y\right]$ such that the model $\mathbf{f}_{\text{clip}}$ maps $X$ to a latent representation $Z = \mathbf{f}_{\text{clip}}(X) \in \mathbb{R}^{F \times T}$. We can approximate the objective by taking the softmax of the dot product of $Z$ and the candidate speech representations $Y_j$:

$$\hat{p}_j = \frac{e^{\langle Z, \tilde{Y}_j \rangle}}{\sum_{j'=1}^{N} e^{\langle Z, \tilde{Y}_{j'} \rangle}}, \tag{3}$$

where $\langle \cdot, \cdot \rangle$ is the inner product over both dimensions of $Z$ and $\tilde{Y}$ Défossez et al. (2023). The CLIP loss is thus the cross-entropy between $p_j$ and $\hat{p}_j$, simplifying to:

$$L_{\text{CLIP}}(p, \hat{p}) = -\log(\hat{p}_N) = -\langle Z, Y \rangle + \log \left( \sum_{j'=1}^{N} e^{\langle Z, \tilde{Y}_{j'} \rangle} \right) \tag{4}$$

under the assumption of a dataset large enough that the probability of sampling the same segment twice can be neglected (Défossez et al., 2023).

### A.4.3 Confusion Loss

The formal definition of the confusion loss as introduced by Tzeng et al. (2015) and used by Dinsdale et al. (2021) is given as:

$$L_{\text{conf}}(X_u, d_u, \Theta_d; \Theta_{\text{repr}}) = -\frac{1}{S_u} \sum_{s=1}^{S_u} \sum_{k=1}^{N} \frac{1}{N} \log(p_{s,k}) \tag{5}$$

where only the parameters in $\Theta_{\text{repr}}$ are updated depending on the fixed value of $\Theta_d$ as indicated by $L_{\text{conf}}(X_u, d_u, \Theta_d; \Theta_{\text{repr}})$.

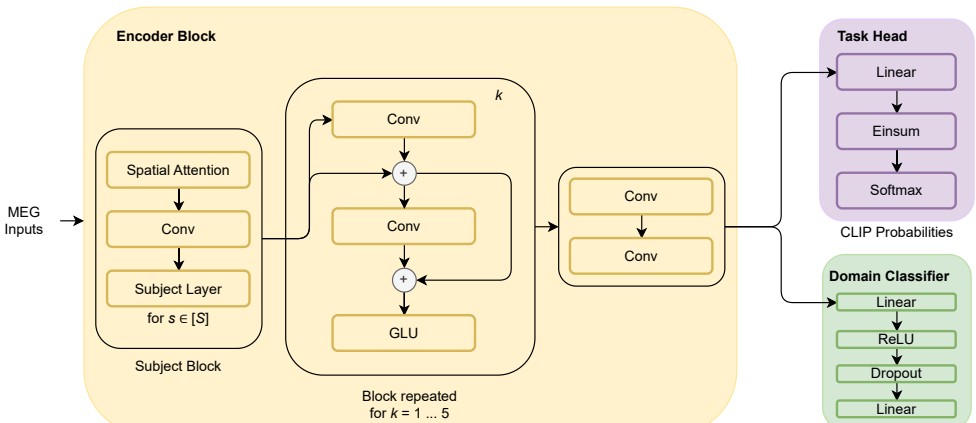

Figure 9: The Brainmagick architecture (Défossez et al., 2023) as modified for feature-level harmonization. No activation functions are used in the subject block. In the five repeating convolutional blocks, the first two convolutions use a residual skip connection, increasing dilation, BatchNorm layer, and GELU activation. The final convolution in these blocks is not residual and halves the number of channels with a GLU activation. The encoder block then applies two $1 \times 1$ convolutions with a GELU activation after the first. We use same domain classifier as Dinsdale et al. (2021), and use the CLIP network for the primary decoding task. The einsum refers to the tensor operation to calculate the normalized similarity scores between candidate and estimate segments.

### A.4.4 DATALOADERS

The original code from Défossez et al. (2023) creates custom classes which track the raw .wav files of the simulus and MEG recordings in blocks chunked by seconds moving forward in time. For the purposes of the current study, we adopt the convention from the original paper and use a 6 second minimum block size. Additionally, because a linguistic representation is used directly in the decoding step, the custom dataclasses also enforce that across all individual subjects the train, validation, and test segments are mutually exclusive with respect to the sentences presented as stimulus. This is maintained for the case where the splits are built from more than one dataset. We further modify this behavior to ensure that data within every batch is tracked with an identifier for its dataset of origin. This information is then extracted at train time to form the ground truth vectors for the domain classifier. Support for the selection of specific subjects, enabling study of subsets of any size and construction, was also added during the completion of the present study.

### A.5 ADDITIONAL MEGALODON DETAILS

Jayalath et al. (2024) define three pretext tasks for speech decoding: band prediction, phase shift prediction, and amplitude scale prediction. The band prediction task randomly selects and applies a band-stop filter to one of the frequency bands typically associated with brain activity: Delta (0.1-4 Hz), Theta (4-8 Hz), Alpha (8-12 Hz), Beta (12-30 Hz), Gamma (30-70 Hz), and High Gamma (¿70 Hz) (Giraud & Poeppel, 2012; Piai et al., 2014; Mai et al., 2016). The goal is to then predict the frequency band which was rejected. The phase shift prediction task is similar in nature: a discrete uniform random phase shift is applied to a uniform randomly selected proportion of the MEG sensors, with the goal of predicting which phase shift was applied (a discrete number of possible values are used in order to reduce the difficulty of the task by treating it as a multi-class problem) (Jayalath et al., 2024). The use of random sensors and uniform random selection is meant to mitigate the effect of variance in sensor placement between studies by ensuring the differences between any two regions of the brain are represented. Finally, the amplitude scale prediction task selects a random proportion of the sensors and applies a discrete random amplitude scaling coefficient to the signal

with the objective of predicting the scaling factor (Jayalath et al., 2024). The intention behind this task is to learn representations encoding relative sensor amplitude differences. These pretext tasks are used to pre-train the backbone, a dataset-conditional layer and encoder block, of the architecture (see fig 10) before being swapped out for the speech decoding tasks in the fine-tuning stage. Additionally, subject conditioning (via subject embeddings, in contrast with Défossez et al. (2023)) is applied at the bottleneck of the encoder block before the pre-text or fine-tuning task heads.

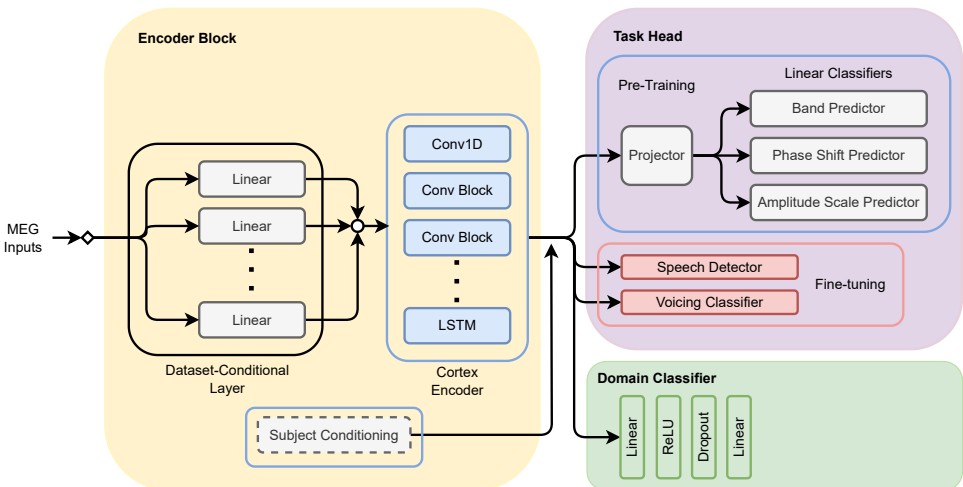

Figure 10: The MEGalodon architecture as modified for feature-level harmonization. Layers treated as part of the encoder block, task head, or domain classifier are respectively shown in yellow, purple, and green. All weights updated during pre-training are shown in blue. Weights trainable during fine-tuning are in red, with the addition of those in the encoder block in the case of deep fine-tuning.

### A.5.1 DATALOADERS

The MEGalodon architecture is already capable of supporting multiple datasets during a single training run, and does this by leveraging the MultiDataloader class from the PNPL python library[7]. Under the original implementation, one batch from each dataset is returned in alternating fashion during training. Adversarial harmonization, however, is better served when every dataset is represented by at least one data-point in each batch. We accomplish this by creating a custom dataloader class, ComboLoader, able to take a list of other dataloaders and return batches in the form of a tuple containing a single batch from each of the original loaders. At train time, random slices, the sum of which is equivalent to the original batch size, are then taken from each batch in this tuple and processed. Upon aggregating, the effect becomes equivalent to that of a batch with the originally specified size but having a random mix of the data from every domain. The ground truth targets for the domain classifier are also calculated at train time using the lengths of the randomly generated batch slices. This approach allows for the implementation of adversarial harmonization with minimal changes to the underlying network architecture. The PNPL datasets additionally support returning metadata alongside each MEG recording, and this feature was used to extract the associated participant IDs which could then be used to retrieve the correct ages when creating the target vectors for age-based harmonization.

### A.6 SOFTWARE CHALLENGES

While the use of popular deep learning libraries such as Pytorch and Lightning has many advantages, it can also lead to unexpected roadblocks when attempting to implement behaviors outside the expected scope of their standard workflows. This was the case for the present work when attempting to build out the functionality for adversarial harmonization. As a reminder, the adversarial phase of

---

[7]https://github.com/neural-processing-lab/pnpl

the harmonization framework we implement is composed of three steps: (1) optimizing the encoder block and task head for the task of interest, (2) optimizing the domain classifier to maximize its ability to identify the target bias, and (3) optimizing the encoder block to erase any signal related to the target bias from its features. This means that every training step during this phase contains three backwards passes and three steps by different optimizers. However, the same initial set of features produced by the encoder block is used across all of these functions. Inevitably, this leads to clashes in the computational graph as the parameters of the encoder block are updated after the first optimizer's step call, but the same feature vector used in the third step is still associated with the previous version of those parameters. In older versions of Pytorch, setting the `retain_graph` parameter to `True` when calling the backward pass successfully navigated this issue. However, this behavior was not maintained for manual optimization in Lightning. Instead, we were compelled to implement a work-around which involved re-writing the forward pass for all layers of the encoder block such that the parameters of that layer are cloned and passed to its respective Pytorch Functional variant alongside the input. It should be noted that slight differences in the underlying construction of these layers from their named counterparts does introduce a numerical variance from that of the original models, however, it is on a order of magnitude small enough that it did not relevantly impact performance. The versions of all tensors related to one training step were examined to ensure that optimization continued to perform as expected when applying this procedure.

## A.7  ADDITIONAL RESULTS

The subset results for the augmented Brainmagick architecture are shown in Table 4. We conduct a one-sided independent samples $t$-test using the subset results collected across three seeds. We find that the effect of adversarial harmonization on top-10 accuracy is statistically significant ($p < 0.05$) when evaluating on both the Gwilliams Gwilliams et al. (2022) test split ($p = 0.0021$) and the MOUS Schoffelen et al. (2019) test split ($p = 0.0358$). This is again demonstrated when training over the full datasets as seen in Table 2 and we include the Top-1 accuracy results here in Table 5 for completeness. As in the Top-10 case, the effect is statistically significant ($p < 0.05$) when evaluating on both the Gwilliams Gwilliams et al. (2022) test split ($p = 0.0163$) and the MOUS Schoffelen et al. (2019) test split ($p = 0.0141$). These results clearly show the ability of adversarial harmonization to enhance deep-learning architectures for cross-dataset generalization of MEG speech decoding where they might otherwise be unable to do so.

| Subset Results | | Top-10 Accuracy | |
|---|---|---|---|
| **Method** | **Training Data** | Gwilliams | MOUS |
| Control | Gwilliams + MOUS | $65.9\% \pm 0.2$ | $57.7\% \pm 0.5$ |
| Harmonized | Gwilliams + MOUS | $\mathbf{67.8\%} \pm 0.2$ | $\mathbf{59.6\%} \pm 0.6$ |

Table 4: As in the full-run case, we report Top-10 segment-level accuracy. The best performance recorded over the test split of each dataset subset is marked in bold. Confidence intervals are calculated over 3 seeds.

| Full-Run Results | | Top-1 Accuracy | |
|---|---|---|---|
| **Method** | **Training Data** | Gwilliams | MOUS |
| Control (Official repo) | Gwilliams | 41.2%, 41.3%* | - |
| Control (Official repo) | MOUS | - | 40.4%, 36.8%* |
| Control (Our implementation) | Gwilliams | 69.8% | - |
| Control (Our implementation) | MOUS | - | 37.8% |
| Pre-trained on MOUS | Gwilliams | 39.4% | - |
| Pre-trained on Gwilliams | MOUS | - | 36.9% |
| Control | Gwilliams + MOUS | $39.2\% \pm 0.5$ | $36.6\% \pm 0.5$ |
| Harmonized | Gwilliams + MOUS | $\mathbf{41.4\%} \pm 0.3$ | $\mathbf{38.8\%} \pm 0.2$ |

Table 5: Top-1 segment-level accuracy with confidence intervals calculated over 3 seeds. Results as reported in the original study are denoted by a single asterisk (*). The best performance recorded over each validation dataset is marked in bold.

| Gwilliams Fine-Tuning | | Balanced Accuracy | |
|---|---|---|---|
| Method | Pre-training Data | Speech Detection | Voicing |
| Control | Balanced Subset | 51.1% | - |
| Control | Random Subset | 50.6% | - |
| Warm-up Only | Balanced Subset | 50.8% | - |

Table 6: We report the augmented MEGalodon architecture balanced accuracy results for the speech detection and voicing classification tasks, fine-tuning and testing with CPU on the Gwilliams dataset. All pre-training conditions are equivalent to those described in Table 3.

## A.8 COMPARISON OF ADVERSARIAL OPTIMIZERS

Here we demonstrate the increased stability provided by choosing SGD over Adam as the optimizer for the domain classifier during adversarial harmonization. In figures 11, 12, and 13, we plot the task performance of the MEGalodon base architecture training on the age-balanced subset for 200 epochs, with the warm-up phase ending at epoch 100. Besides the choice of optimizer, all other hyperparameters are held equal.

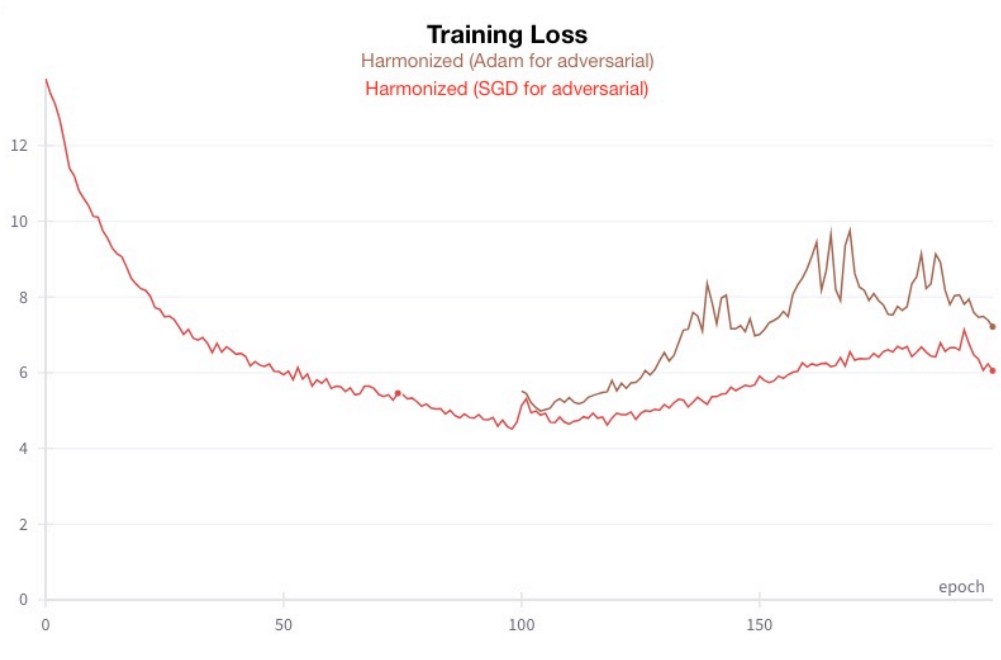

Figure 11

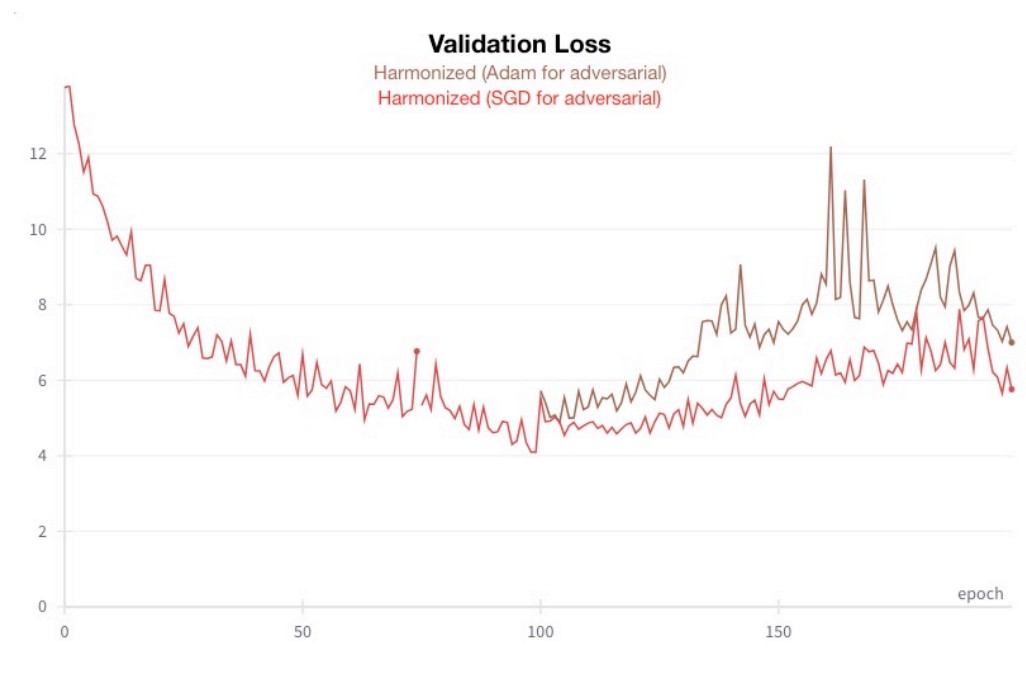

Figure 12

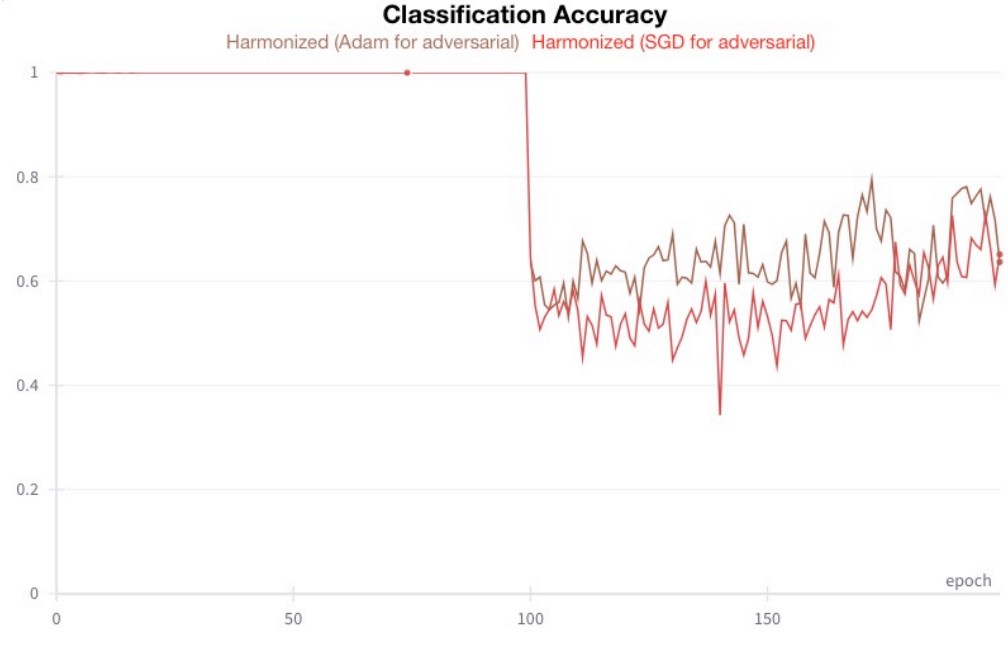

Figure 13

## A.9 ANALYSIS OF TRAINING DURATIONS

Below in figures 14 and 15 we show that, holding all else equal, beginning the harmonization phase later into training does not mitigate the tendency of the task loss to diverge. The following plots do, however, demonstrate that after an initial peak following the start of harmonization, the task loss once again begins to trend downwards. Note that convergence (as determined by early stopping)

is not reached by the control even within 400 epochs, which is why we opt for an epoch-based scheduling of the warm-up phase in the case of MEGalodon.

Figures 16 and 17 demonstrate the case of training on smaller batch sizes. Overall loss is reduced, but the model still fails to reach convergence within 200 epochs and thus beginning harmonization during this time still leads to divergence.

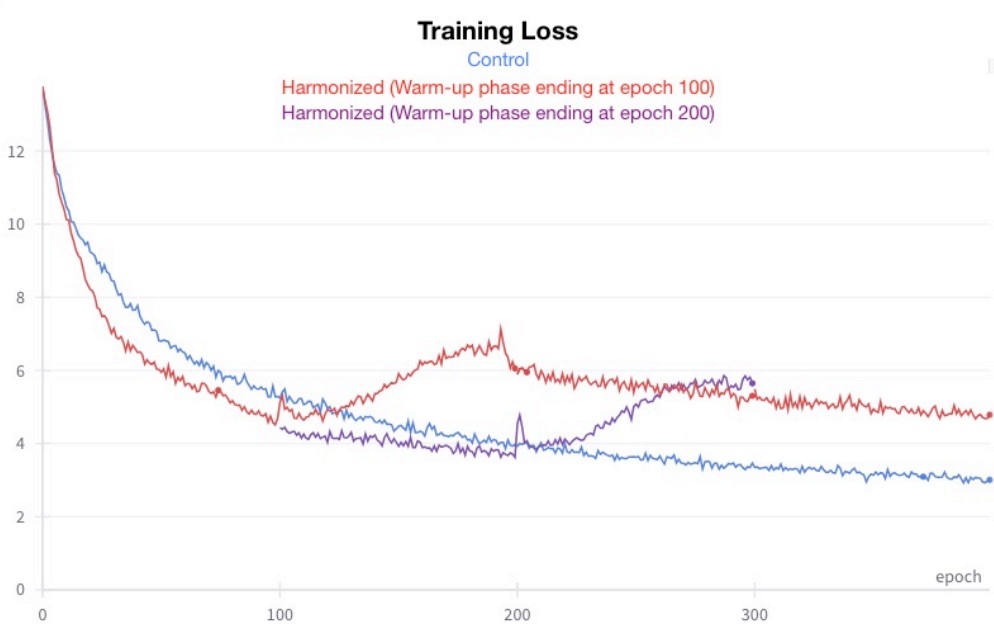

Figure 14

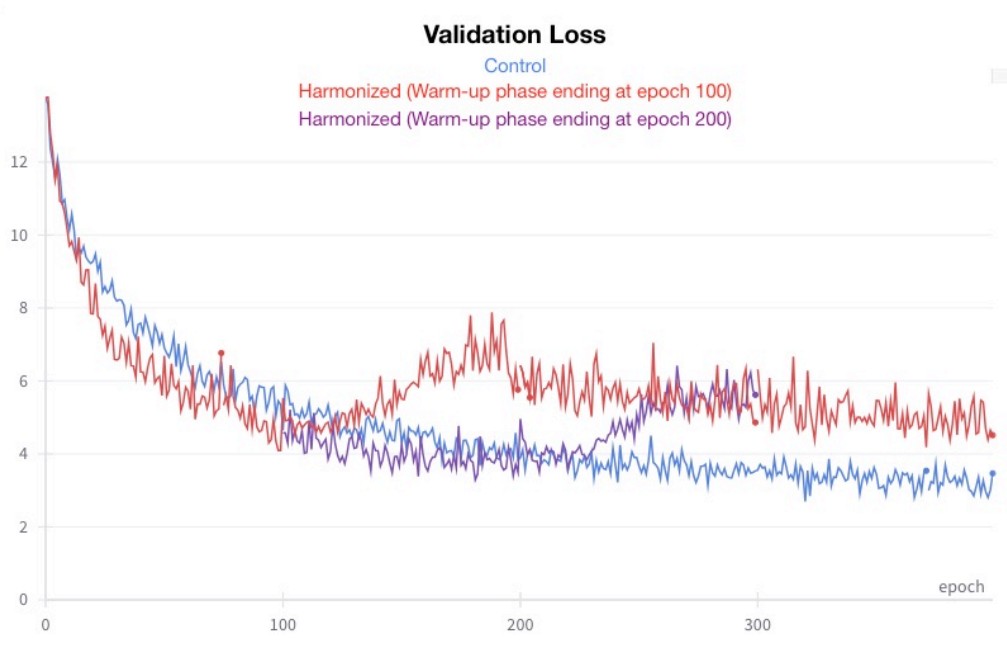

Figure 15

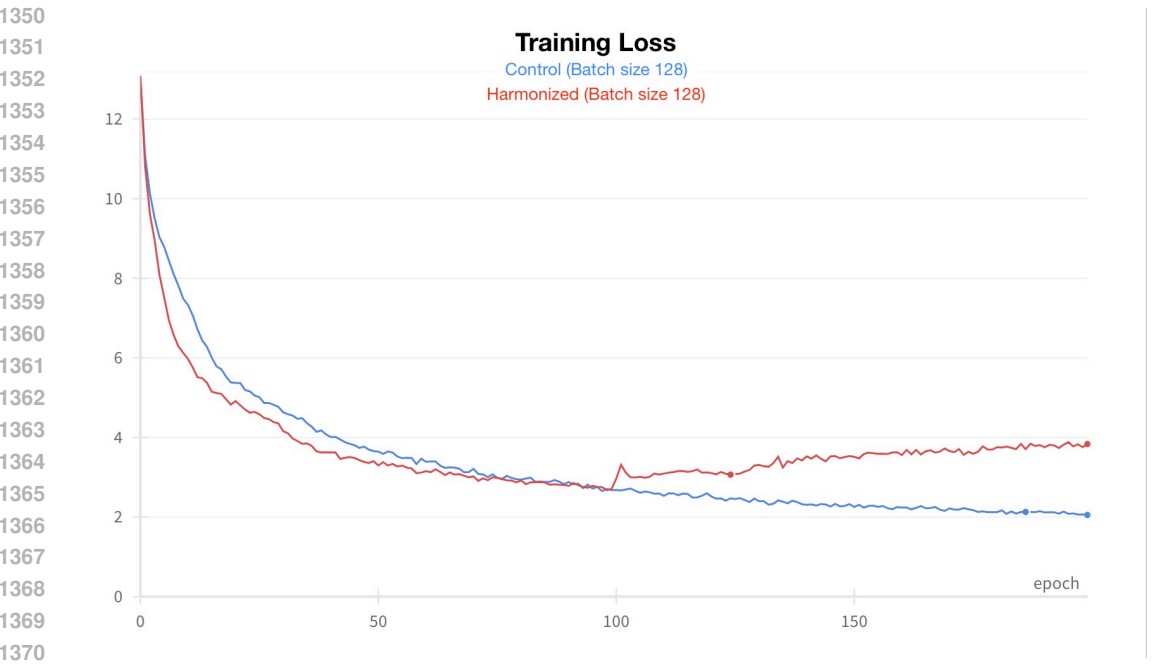

Figure 16

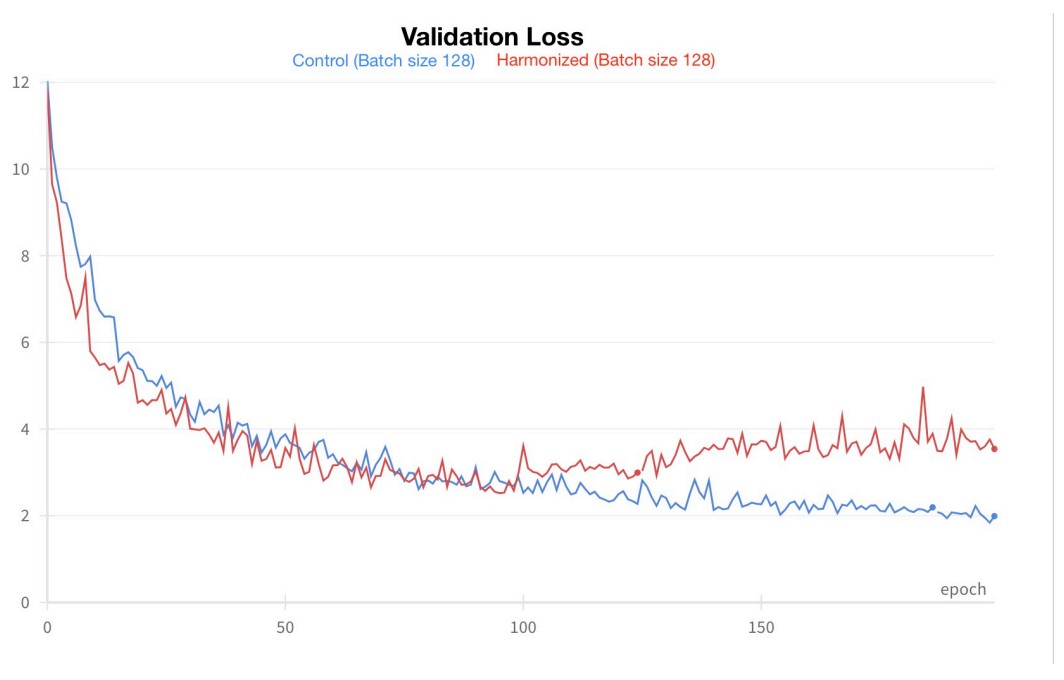

Figure 17

## A.10 HYPERPARAMETERS

| Parameter | Value | Final Validation Loss |
|---|---|---|
| Harmonization phase start | Epoch 25 | 7.61 |
| Harmonization phase start | Epoch 100 | 5.76 |
| Alpha | 1.0 | 5.76 |
| Alpha | 0.33 | 7.30 |
| Alpha | 0.25 | 5.65 |
| Optimizer | Adam | 7.00 |
| Harmonization phase LR | 0.000001 | 5.76 |
| Harmonization phase LR | 0.000005 | 5.88 |
| Harmonization phase LR | 0.000066 | 5.98 |

Table 7: Hyperparameter testing of the augmented MEGalodon architecture carried out over subsets of the MOUS and Cam-CAN datasets. Tested parameters are listed, with all other values for that run held constant. Validation loss is reported at epoch 200. The final configuration of the model used for running experiments is an alpha of 0.25, beta of 1 (choice of beta value was negligible), harmonization phase start of 100, and harmonization learning rate of 0.00002 for the task and classifier optimizers and 0.00001 for the adversarial optimizer. The choice to set the adversarial learning rate to half that of the others came recommended by Dinsdale et al. (2021) to increase training stability.

## A.11 LIMITATIONS

The present study was limited by time and resource constraints which ultimately meant results could not be collected across multiple seeds in all cases. Additionally, testing over the full datasets was not carried out for the experiments using the MEGalodon Jayalath et al. (2024) base architecture. Given the flexibility of the chosen harmonization framework, the present study would have benefited from exploring its capacity to combine more than two datasets at a time. An initial look at pooling three datasets for pre-training was done but not investigated further within the scope of this work. A significant bug in the code was discovered relatively late into the project which forced the experimental results collected up to that point to have to be discarded and re-collected. This setback meant that more extensive testing of the kind discussed above was infeasible. While this is regrettable, bugs in large and complex codebases, particularly those that build on other's publicly available code, can be commonplace. It is important to us to have caught the bug and be able to present accurate results, rather that allow it to remain undiscovered. An extension of the framework to enable harmonization of datasets with skewed demographic biases, such as the age distributions of MOUS Schoffelen et al. (2019) and Cam-CAN Shafto et al. (2014); Taylor et al. (2017), is noted in Dinsdale et al. Dinsdale et al. (2021). In this variant, the datasets are trained on in full, but harmonization is only carried out using subjects from the overlapping area of the distributions. This extension was implemented but was not able to be properly tested at the current time.

