# OpenReview forum: "Resolving Domain Shift For Representations Of Speech In Non-Invasive Brain Recordings"
_ICLR.cc/2025/Conference — Submitted to ICLR 2025_

### Official Review · Reviewer_EzYE · 2024-11-03

**Soundness:** 1
**Presentation:** 2
**Contribution:** 1
**Rating:** 3
**Confidence:** 4

**Summary:**

The authors attempted to demonstrate that the use of standard feature based adversarial domain shift adaptation machinery allows for  an improved generalization across time-resolved functional MEG based brain imaging datasets. They used two pairs  of MEG datasets reported in previous studies and recorded from multiple participants listening to audio stories. The authors augmented the two *existing*  DL models Brainmagick and MEGalodon with an *existing* adversarial domain shift adaptation strategy (Dinsdale et al., 2021) and showed that harmonizing the datasets based on the participant's age slightly improves the classification performance in the downstream task in the first (Gwilliams-MOUS) pair of datasets where the goal was to predict the index of a 3-s long audio segment.  The authors then put forward the claim that harmonizing with respect to participant's age is an important step in dealing with multi-subject MEG datasets.

**Strengths:**

1. The authors (although they are not the first ones)  address the important problem of aligning the brain imaging data from multiple subjects \ clinics\ devices.
2. The authors use state-of-the-art multi-subject  MEG datatasets
3. The authors successfully implemented  the existing MEG decoding solutions and reproduced the results from the original papers.

**Weaknesses:**

1. **Utility.** First of all, I am concerned with the overall utility of this study. The use of brain imaging methods to decode parameters of the external auditory stimuli does not bring us any closer to the development of speech prosthesis device where the DL-based decoding of brain's electrical activity appears really useful. See for example  https://www.biorxiv.org/content/10.1101/2024.08.21.608927v1 where decoding of the imagined (covert) speech is done at the chance level by the architecture capable to successfully recover both the perceived and the overt speech. The study uses invasive macroscale recordings but the situation with non-invasive imaging methods is even worse. The only real speech-BCI solutions existing to date (see the studies by Stavitsky's lab @ UC Davis) are based on the brain activity data recorded  with  intracortical probes (Utah arrays) that sample activity of individual neurons or their very small populations.  These data are really uniquely informative and can be used to decode the attempted speech (https://www.medrxiv.org/content/10.1101/2023.12.26.23300110v1).
2. **Approach justification.** At the same time drastically simpler models (comprising only tens of parameter) perform well on the decoding of the external stimulus parameters from brain activity (SPOC, Nikulin et al) just because brain's reaction is quite strong to such an external stimulation. Combined with simple domain shift adaptation techniques such as https://arxiv.org/abs/2407.03878 or earlier approaches, say, https://ieeexplore.ieee.org/document/8624413 these simpler techniques need to be brought into the scene of comparison.  If the authors want to "untrain" w,r.t to age feature they could have pooled the dataset withing age strata and perform the described  stable and well define domain adaptation methods. At the very least the proposed approach has to be leveraged against these more stable techniques,
3. **Clarity.** The paper is quite poorly written. It has a lot of unnecessary and unrelated to the main idea  comments and It was not easy to understand what the authors actually decided to do. Please, also see the questions. From what I managed to deduce the work is rather incremental and does not represent a significant advance neither in the magnitude of the achieved improvements, nor methodologically.
4.  **Data management\sanity.** If deduced correctly, the authors used 15% of the datasets for their experiments which  raises concerns regarding the reliability of the reported results and the observed minimal improvements. While I am sympathetic with the described shortage of computational resources these arguments are hardly acceptable  as a justification especially given that the MEG datasets used are already not so large. Please, also see the questions.
5. **Impact.** Please, see the questions below, but the overall results are rather inconclusive and hardly support the claims made in the paper regarding the effect of datasets harmonization based on the age parameter as this statement can be made (and with a slight stretch) only based on the Gwilliams-MOUS dataset analysis results pair and not on those for the second pair. Leave along the lack of confidence intervals in Table 3.  Also the age is not the strongest source of the between-subject MEG variability yet it can be predicted in the 25~85 y.o. range with more or less uniformly distributed accuracy from the non-invasive recordings of brain activity  https://www.sciencedirect.com/science/article/pii/S105381192200636X.
6. **A suggestion for increasing the impact and reliability** Age prediction from younger people is typically more accurate which may be an interesting avenue for the authors to explore in the future and compare the effect of harmonization for  the young vs. middle-age participants.

**Questions:**

1. Could the authors be more explicit as to what is exactly done to a pair of datasets when dataset A is used for the pre-train and dataset B is used as a new domain dataset. Was the actual downstream  task training (combined with harmonization) performed in this dataset B ? A processing flow diagram would be very helpful to clarify this!
2.  For Gwiliams dataset and using the model from (Defossez, 2023) the authors reported only top-10 accuracy. In the original paper both Top-1 and top-10 are reported. What about the Top-1?
3. How many runs were used to estimate the confidence intervals on the accuracy metrics reported in Tables 2?  Why are these missing in table 3. Control accuracy of 52.60 and the highlighted value of 52.65 seem to be not very different.
4, What do the authors mean under "We do not find that the base Brainmagick architecture is effective in cross-dataset generalization."  - why does the subject specific layer appeared ineffective in translating between the data recorded with KIT and CTF systems? What are the  MEG sensor types used by these two systems?
5. Have the authors tried to prune the explored architectures to combat the computational load? From reading the original papers I have gotten an impression that both Brainmagick and MEGalodon architectures are significantly over-parameterized especially given the amount of training data available. At the same time the parameters (e.g. loudness) of the audio stimulus could be reliably decoded with several tens of coefficients (SPOC, Nikulin et al.) also the electrical activity can be predicted from the audio envelope and vice-versa with significantly above chance accuracy using a single convolutions kernel (https://www.frontiersin.org/journals/human-neuroscience/articles/10.3389/fnhum.2016.00604/full) even from the EEG data.  The two latter approaches use total of 50 to 300 weights to solve their tasks.

---

> ### Author Response · Authors · 2024-11-15
>
> Weaknesses:
>
> Utility - It seems as if the utility of non-invasive decoding studies as a whole is being called into question, though please do clarify your original statement if this is being misinterpreted. Given the high practical and ethical thresholds for surgery, there is certainly a space for studies that are not dependent on invasive techniques. Non-invasive decoding problems are well established in the literature as valuable scientific contributions. As just one example, see the paper by Défossez et al., whose architecture is one of the focuses of this work.
>
> Approach Justification - The call for a comparison with other DA methods would certainly strengthen the paper and is a valuable suggestion. As we have responded to other reviewers, these other methods have yet to be applied to MEG and there remains a substantive difference between statistical transformations in sensor space and the kind of feature-level DA we employ here.
>
> Clarity - We are sorry that you feel the paper is poorly written and would appreciate any suggestions you might have as to what specifically you felt was confusing or unnecessary. With respect to the achieved advancements, we would like to underscore that while the improvements over the original baselines (i.e. architectures training and evaluating on the same dataset) may seem incremental they are (a) statistically significant and (b) much larger improvements when comparing the performance pooling across studies with DA to that of the original architectures when pooling without. We acknowledge that we could do more to emphasize this last point and will make edits to the relevant sections of the paper accordingly.
>
> Data Management - Results for the experiments with the Brainmagick base architecture are reported for both subsets using 15% of the data as well as the datasets in their entirety. While these datasets may seem small in comparison to other domains, this study is engaging with a volume of MEG data larger than the vast majority of other studies in the field (which usually use a single MEG study such as Gwilliams, alone). The experiments conducted with the MEGalodon base architecture were restricted to only the subsets, but it is reasonable to believe the scaling effect demonstrated in the Brainmagick case would hold here as well.
>
> Impact - The primary objective of this paper is to demonstrate the viability of feature-level DA as a path towards integrating more MEG data sources for training a single architecture, and would like to re-emphasize the points made in response to “Clarity”. While we acknowledge your arguments with respect to the age-related analysis, those findings are secondary to the principle impact of this work.
>
> Questions:
>
> With respect to the first question, to which set of experiments are you referring? For Brainmagick, if referring to Table 2 the “Pre-trained on A” simply means that the training of the architecture on B was initialized from the final saved weights of training the baseline on A, as opposed to a random initialization. For MEGalodon, the framework has a pre-training stage and a fine-tuning stage where the dataset in the fine-tuning stage is never used during pre-training. A flow diagram is a good suggestion and could be added to further support the information included in sections A.4 and A.5 of the appendix.
>
> The primary metric highlighted by the original authors is top-10 accuracy, whose convention we follow. For the same reason as Defossez et al. we did not want to spend unnecessary space reporting the top-1 accuracy, though we acknowledge your point and can include those results in the appendix upon revision.
>
> All confidence intervals are computed over 3 seeds as reported in Table 2’s caption. Computational and time constraints prevented additional seeds from being collected for table 3.
>
> "We do not find that the base Brainmagick architecture is effective in cross-dataset generalization" means that the original architecture, without augmentation for DA, has a decline in performance when pooling datasets for training. In other words, without the augmentations applied in this work the original architecture is unable to generalize over multiple studies to improve decoding performance. These results are provided in Table 2.
>
> Reducing parameterization was not the focus of our study, but rather DA and improving cross-dataset generalization. As a result we sought to maintain the two original architectures we augmented as they were proposed. Interesting point though and we will consider this in the future.

---

> > ### Author Response · Authors · 2024-11-25
> >
> > Dear Reviewer EzYE,
> >
> > As the the ICLR 2025 discussion phase is soon coming to a close, we hope to hear from you again. We have addressed each of the points from your initial review in our rebuttal and would be interested in hearing any continuing concerns that you may have. Your input is invaluable in refining our work to be the best it can be.
> >
> >
> > Thank you,
> >
> > The Authors

---

> > ### Comment · Reviewer_EzYE · 2024-11-27
> > **I will keep the score**
> >
> > Thanks! The authors made an attempt to respond  but addressed my concerns only partly and made some incorrect and non-insightful statements. The question regarding  the comparison with much simpler models is answered incorrectly. SPOC was applied to MEG data numerous times: https://mne.tools/stable/auto_examples/decoding/decoding_spoc_CMC.html, https://www.biorxiv.org/content/10.1101/2021.04.23.441088v1.full ,...  The answer about the rationale lacks insights.   Also, as I stated earlier I did not see a clear improvement rendered by the approach proposed by the authors.  The answer regarding 10% Top as a measure of accuracy in unsatisfactory. What about Top-1? My original score is already low enough, I will keep it and given that the authors did attempt to respond I will not lower it further.

---

> ### Author Response · Authors · 2024-11-28
>
> Thank you for your responses, and I am just following up to gather feedback for future iterations of this work. With respect to Top-1 accuracy, I have uploaded a revised copy of this paper with the table included in the appendix (as Défossez et al. did in their paper) and will be included the results at the bottom of this comment. As stated before, these results are in line with what we found for the top-10 accuracy for the full dataset testing and subset testing.
>
> I agree that the paper would be strengthened by additional DA comparisons, though I am still somewhat unclear as to why you feel that the use of current SOTA decoding models as the baselines to be augmented for DA is a worse choice than augmenting a simpler architecture. Both references you include (the second of which is not a peer reviewed paper) for SPOC, while used with MEG data, are not examples of an application to a speech decoding task. I feel it may be worth clarifying at this time, as I have also done in the revised copy of this paper, that the speech detection task for MEGalodon in this work is not the identification of the onset of speech from resting state MEG but rather the detection of natural breaks between words from an audio stimulus of continuous speech. This is a significantly more challenging task than classifying the onset of auditory stimulus.
>
>
> | **Method**                          | **Training Data**    | **Gwilliams**         | **MOUS**             |
> |-------------------------------------|----------------------|-----------------------|----------------------|
> | **Full-Run Results**                |                      | **Top-1 Accuracy**    |                      |
> |                                     |                      | Gwilliams             | MOUS                 |
> |-------------------------------------|----------------------|-----------------------|----------------------|
> | Control (Official repo)             | Gwilliams            | 41.2%, 41.3%*         | -                    |
> | Control (Official repo)             | MOUS                 | -                     | 40.4%, 36.8%*        |
> | Control (Our implementation)        | Gwilliams            | 69.8%                 | -                    |
> | Control (Our implementation)        | MOUS                 | -                     | 37.8%                |
> | Pre-trained on MOUS                 | Gwilliams            | 39.4%                 | -                    |
> | Pre-trained on Gwilliams            | MOUS                 | -                     | 36.9%                |
> | Control                             | Gwilliams + MOUS     | 39.2% ± 0.5       | 36.6% ± 0.5       |
> | Harmonized                          | Gwilliams + MOUS     | 41.4% ± 0.3       | 38.8% ± 0.2       |

---

> > ### Comment · Reviewer_EzYE · 2024-11-28
> >
> > I would not stress the difference between EEG and MEG data. Conceptually and structurally these kinds of time resolved neuroimaging data are very similar. Defossez applies the same architecture to both types of data. At the same time here is another peer reviewed work (of which there are many ) https://www.sciencedirect.com/science/article/pii/S1053811920303797 showing the use of SPOC and other VERY simple models for solving several downstream tasks.
> >
> > The tasks these models are used for are indeed much simpler than the decoding of the perceived 3-s long segments but I used SPOC as a limiting example of a VERY simple model  with the number of parameters equal to the number of channels
> > (100-300) in the data.  My own experiments show that Defossez's architecture can be significantly shrunk and yet maintains the performance and even has it slightly improved.  I thought that simpler models could have benefited more from the  harmonization approach you are proposing. My intuition here is supported by the fact that in Defossez's model the subject adaptation layer ( working towards a similar goal as your harmonization) contributes very significantly to the improved performance, see their ablation study table.

---

> > > ### Author Response · Authors · 2024-12-01
> > >
> > > Thank you for your feedback with respect to MEG/EEG, and in future work we will certainly expand our comparison methods to cover those primarily applied to EEG.
> > >
> > > Your feedback with respect to model complexity is very helpful as well, and we agree that we might be able to elicit a stronger effect on overall performance when using more streamlined models. However, in our view, the primary goal of this work was to demonstrate the ability of this framework to induce cross-dataset generalization in architectures where it did not exist before. To that end, we believe this work was successful and the protests about the selection of base architectures to be tangential. In your opinion, what more should be done that would convince you about the performance of the primary objective? Just broadening the array of comparison harmonization methods?
> > >
> > > We greatly appreciate your continued engagement.

---

### Official Review · Reviewer_cg3A · 2024-11-03

**Soundness:** 2
**Presentation:** 3
**Contribution:** 2
**Rating:** 3
**Confidence:** 4

**Summary:**

This paper investigates the use of non-invasive EEG imaging techniques, specifically magnetoencephalography (MEG), to decode speech representations in the brain. Aiming at the problem of poor generalization ability of non-invasive brain imaging data across multiple datasets, the authors adopt an adversarial domain adaptation framework to improve the model generalization between different datasets. In this study, the authors leverage two different speech decoding models and reconcile the differences between datasets through a feature-level adversarial learning approach, which "obfuscates" the domain classifier through an enhanced network architecture to reduce the impact of dataset bias.

**Strengths:**

1. This study is the first to implement feature-based adversarial domain adaptation on MEG data, which provides a new methodology for cross-dataset application of non-invasive EEG imaging techniques.
2. By improving the generalization ability of non-invasive neuroimaging technology, it is helpful to improve the accuracy and reliability of speech decoding, which is of positive significance for practical applications such as rehabilitation of speech disorders.

**Weaknesses:**

1. The paper does not describe its own method in detail, especially when most of the model methods are built on the basis of others, which will weaken the innovation of the paper.
2. The experimental analysis of the paper is seriously insufficient, and some statistical charts take up too much space, resulting in less practical useful length of the paper.
3. The overall writing of the paper is more like an experimental analysis report than a complete paper.

**Questions:**

1. The authors show that after 100 epochs, the distribution tends to the average distribution of all age groups, however, it is not obvious in Figure 4, the author should provide the trend of the distribution with epochs to prove this.

---

> ### Author Response · Authors · 2024-11-15
>
> Weaknesses:
>
> We strived to be as transparent as possible with our methods. If you could share which specific aspects of the methodology you were unclear on, these details can be highlighted in further edits. We also acknowledge that the primary contribution of our work is their application to a previously unexplored modality and the related insights with respect to MEG DA, rather than proposing an entirely new framework. The objective was not to introduce a novel theoretical approach to DA.
>
> Questions:
>
> This is a good suggestion and such a formatting would have been a stronger visualization of the effect being shown. Unfortunately, the model was only checkpointed every 100 epochs, meaning we cannot generate such a series of graphics at this time. We opted instead with Figure 4 to contrast the probability distributions before and after DA. Would it be more illustrative to provide a contrast with the probability distribution after 200 epochs without applying DA as well?

---

> > ### Author Response · Authors · 2024-11-25
> >
> > Dear Reviewer cg3A,
> >
> > As the the ICLR 2025 discussion phase is soon coming to a close, we hope to hear from you again. We have responded to the points from your initial review in our rebuttal and would be interested in hearing any continuing concerns that you may have. Your input is invaluable in refining our work to be the best it can be.
> >
> > Thank you,
> >
> > The Authors

---

### Official Review · Reviewer_hbhB · 2024-11-03

**Soundness:** 3
**Presentation:** 2
**Contribution:** 2
**Rating:** 5
**Confidence:** 3

**Summary:**

The paper proposes a domain adaptation (DA) method that leverages adversarial training techniques to enhance generalization across different neural recording datasets. The study evaluates the effectiveness of the proposed framework on two model architectures for MEG (magnetoencephalography) classification tasks related to speech. Results demonstrate that the DA framework can effectively harmonize feature representations across different datasets, particularly between those with distinct demographic characteristics. Additionally, classification accuracy is improved for certain tasks.

**Strengths:**

1. DA is an important topic, especially for neuroscience-related tasks, due to the high variability and limited data volume in this field. This paper could have a notable impact as one of the efforts to apply DA to MEG data.
2. The authors provide an implementation of an existing model, promoting openness and reproducibility in the community.
3. Being the first to successfully apply a similar DA framework in computer vision to MEG field and enable a more general DA for MEG that can be applied on top of different architecture.

**Weaknesses:**

The related work section lacks an adequate discussion on DA methods for EEG. Numerous studies have applied DA to EEG, and given that EEG and MEG share many properties, methods developed for EEG should, in principle, be applicable to MEG and could serve as benchmarks. Including a comparison with a benchmark DA method would enhance the quality of this work significantly.

**Questions:**

1. Could you clarify the number of samples N used in the statistical tests for Table 2?
2. The contribution of the methods section needs further clarification. What are the key contributions and differences between your proposed framework and existing adversarial harmonization procedures?

---

> ### Author Response · Authors · 2024-11-15
>
> Weaknesses:
>
> A comparison with DA methods for EEG was omitted at this time as the current focus was on MEG, a modality that has been largely overlooked with respect to DA thus far. Granted, EEG and MEG do share many similarities but they remain distinct data sources.
>
> Questions:
>
> All confidence intervals and statistical power analyses were conducted with three seeds per assay. The key contributions of the paper are a novel application of feature-level DA to MEG data. This is important as this approach promotes minimal feature engineering of the raw data, avoiding potential sources of bias or accidentally removed signal. While this framework has been used before for computer vision and 3D neuroimaging data (MRI), this is the first application of it to time-series neuroimaging data and demonstrates its potential as an architecture-agnostic approach towards pooling multiple studies. In a functional sense, we also contribute findings supporting the importance of optimizer choice with respect to ADDA frameworks (e.g. improvements using SGD over ADAM for the adversarial head). Other DA methods for time-series neuroimaging data (EEG) are established in the literature, but are grounded in statistical transformations of the input data rather than harmonization at the model level.

---

> > ### Comment · Reviewer_hbhB · 2024-11-23
> >
> > Thank you for the explanation. I can better appreciate your contribution and how it could be impactful to the field. However, I still think including a comparison with a benchmark DA method would enhance the quality and the contribution of this work significantly.

---

> > > ### Author Response · Authors · 2024-11-25
> > >
> > > We would like to thank the reviewer for their thoughtful and continued engagement with our work. We will certainly follow through on your suggestion in future work stemming from this paper.

---

### Official Review · Reviewer_CkVM · 2024-11-04

**Soundness:** 3
**Presentation:** 2
**Contribution:** 2
**Rating:** 5
**Confidence:** 4

**Summary:**

Authors present the first deep MEG dataset alignment/adaptation method, albeit a common goal outside of the deep learning methods and MEG modality. For instance, [1] apply MEG dataset alignment based on canonical correlation analysis and [2] apply deep domain adaptation on fMRI datasets.

They follow a procedure similar to the adversarial discriminative domain adaptation (ADDA) method, that is inspired by the H-Divergence theorem. This procedure, referred as adversarial harmonization, is adopted from a previous study by Dinsdale et. al.. The goal is to infer the domain-specific signal and penalize it to pass a domain-general signal across-datasets.

They utilize two different base architectures in the experiments, from MEGalodon and BrainMagick studies, where a major contribution claim is the re-implementation of BrainMagick base architecture.  MEGalodon base model experiments are on MOUS and Cam-CAN datasets, whereas BrainMagick experiments are on MEG-MASC and Gwilliams datasets. BrainMagick decoding task is the estimation of speech stimulus segment in a given interval. MEGalodon decoding tasks are determining speech occured in a given interval and classification of the phoneme into voiced of voiceless in the MEG data interval. Additionally, there are pre-text tasks in the MEGalodon; namely band, shift and amplitude scale prediction tasks.

Network has three components: encoder, task head, domain classifier. The Components are trained together in a warmup phase, followed by the adversarial parts, the domain classifier optimization to identify dataset bias, and the removal of the dataset bias by confusing the domain classifier.

Adversarial harmonization results are close to the official implementation results of each external study. They emphasize the role of age related variation to be the most significant source of divergence in the data distributions. Authors discuss unstability issues regarding the adversarial harmonization.

[1] Q. Zhang, J. P. Borst, R. E. Kass, and J. R. Anderson, “Inter-subject alignment of MEG datasets in a common representational space,” Human Brain Mapping, vol. 38, no. 9, pp. 4287–4301, 2017, doi: 10.1002/hbm.23689.

[2] J. Wolleb et al., “Learn to Ignore: Domain Adaptation for Multi-Site MRI Analysis,” Jun. 07, 2022, arXiv: arXiv:2110.06803. Accessed: Nov. 03, 2024. [Online]. Available: http://arxiv.org/abs/2110.06803

**Strengths:**

Authors suggest the first adversarial domain adaptation method on MEG modality. The study re-implements the previous work, BrainMagick, base model architecture. Implementing on new modalities requires a tedious work, hence the open source re-implementation of the base model is a strength of the study.

**Weaknesses:**

1- Adversarial harmonization and experiment sections can benefit from a more clear organization. The narrative writing style is easy-to-read in the methods section, however many important parts are in the appendix sections, most probably due to the page limit. A more compact re-writing that includes important information like training details, model figures and training curves, would improve the flow of the paper.

2- Subject level differences are not addressed in the harmonization method. On the similar functional neuroimaging modalities, like EEG (sensor-space) and fMRI (brain-space), similar adversarial domain adaptation methods do not stand out as the best option, as the variation across samples depend on subject-specific differences and dataset/acquisition-site differences, as well as on the demographics. Hence, applying subject level harmonization along with dataset and demographic level harmonization might improve the results. See the questions section for a similar study/method.

3- In both table 2 and table 3, the performance improvement do not support the positive impact of adversarial harmonization in a clear way.
In table 2, control results for both Gwilliams and MOUS datasets are in the mean±std range of Harmonized results.
In table 3, for speech detection, "Harmonized" methods perform worse, and the "Warm-up Only" introduces 0.5% accuracy improvement, whereas for Voicing, "Harmonized" method introduces 0.05% improvement. This suggests the increase in performance might be depending on the number of epochs, a parameter that is not inferred from data and may have a different effect for different datasets/subjects. Hence, the stability of the adversarial harmonization method is not convincing.

**Questions:**

A question regarding a simple baseline: While deep learning has a great impact in many fields, domain adaptation/generalization methods are quite tricky [1]. There are also lightweight methods that focus on inter-subject alignment of MEG data, for instance the work of Zhang et. al. [2] that implements a hyperalignment method. Would it be possible to add a similar method as a baseline? This can benefit the completeness of the results, such that an estimation of a simple spatial transformation, i.e. Procrustean Transform, applied in sensor space can show the relative impact of the Harmonization method.

[1] I. Gulrajani and D. Lopez-Paz, “In Search of Lost Domain Generalization,” presented at the International Conference on Learning Representations, Oct. 2020. Accessed: Nov. 05, 2024. [Online]. Available: https://openreview.net/forum?id=lQdXeXDoWtI

[2] Q. Zhang, J. P. Borst, R. E. Kass, and J. R. Anderson, “Inter-subject alignment of MEG datasets in a common representational space,” Human Brain Mapping, vol. 38, no. 9, pp. 4287–4301, 2017, doi: 10.1002/hbm.23689.

---

> ### Author Response · Authors · 2024-11-15
>
> Weaknesses:
>
> 1 - Good suggestions which we will take into account.
>
> 2 - Subject level differences were not addressed because each of the architectures examined already achieved inter-subject generalizability (within a single study). This was done with a subject-specific embedding for MEGalodon and a subject-specific layer for Brainmagick
>
> 3 - The results section for MEGalodon and the discussion addresses the discrepancy between speech detection and voicing and explains why it is an artifact of the original framework’s training protocol. The focus should be on the improvement achieved in voicing classification, the “deep fine-tuning” case. We maintained the original frameworks of the two base architectures we used in order to be as fair as possible while benchmarking. Can you provide more explanation behind the comment on the performance being related to the number of epochs? All benchmark comparisons are trained under the same conditions.
>
> Questions:
>
> This is a reasonable suggestion, though we would like to highlight that one of the benefits of our chosen approach is that by operating in feature space one could apply the demonstrated framework on top of any sensor space transformations (such as Zhang et al.’s MCCA) to likely yield further improvements. We acknowledge the potential concern that the performance gains could be marginal, however, as previous DA studies have failed to demonstrate inter-study generalization with MEG data we maintain that our contributions here are impactful to the field.

---

> > ### Comment · Reviewer_CkVM · 2024-11-21
> >
> > 2- Although each model handles inter-subject variability separately to some extent, harmonization procedure may still introduce a subject-dependent bias. A simple mechanism towards this goal might be to adopt a subject ID classifier akin to a domain classifier for subject bias unlearning, although this will introduce further difficulties in the stability of the optimization phase.
> >
> > 3- My comment is about the scheduling of warm-up and harmonization phases being dependent on the number of epochs (Table 3). If possible, finding a feature/loss dependent mechanism or a context/dataset-independent setting to end the warm up phase might better support the hypothesis of the study.

---

> > > ### Author Response · Authors · 2024-11-25
> > >
> > > We would like to thank the reviewer for their thoughtful and continued engagement with our work.
> > >
> > > 2 - While it is possible that a subject-dependent bias could be introduced, we had little reason to suspect that the harmonization procedure was negatively impacting the existing subject conditioning of the baseline models. Without additional evidence we chose not to prioritize investigating that assumption within the scope of the current study. Additionally, in the MEGalodon framework the datasets (and thus subjects) used during pretraining (where harmonization was applied) were entirely independent of those used for fine-tuning and producing the performance metrics. As performance was improved in both the MEGalodon and Brainmagick instances, if subject-dependent bias was being introduced then the harmonization framework succeeds in improving cross-dataset generalization in spite of this. Future work can confirm this as the framework can support multiple classifiers for different potential confounds, as shown by the combination of dataset and age harmonization in the present study.
> > >
> > > 3 - The goal is to not have the scheduling be dependent on epochs but rather convergence of the task validation loss (as selected by early stopping). This was done in the case of Brainmagick, but proved unrealistic for MEGalodon. Given that convergence was not achieved even within 400 epochs (see figures 14 and 15 in section A.9), we decided to institute the current epoch-based scheduling as we were still able to demonstrate the desired effect while respecting computational limitations. We appreciate you highlighting the potential confusion in our current iteration of this paper and will expand on the appendix discussion as well as address this point more clearly in the main text body.

---

### Meta-Review · Area_Chair_eyzk · 2024-12-12

**Metareview:**

This submission contributes adverserial MEG adaptation methods. The work generated interest and discussion with the reviewers. However, it fail to convince that it met the high bar for acceptance at ICLR. In particular, the reviewers felt that there was insufficient positioning with regards to the large literature on domain adaptation for MEEG, where some dedicated methods for MEEG are well established in the field. In addition, the empirical validation was seen as a bit light.

**Additional Comments On Reviewer Discussion:**

There was a discussion with some good exchanges. But it was incomplete, as one reviewer did not reply.

The discussion did not lead to major changes in positions. Some of the important points discussed where the decoding methods for MEG and EEG.

---

### Decision · Program_Chairs · 2025-01-22

Reject